# Human thymopoiesis produces polyspecific CD8+ α/β T cells responding to multiple viral antigens

Valentin Quiniou[1,2§], Pierre Barennes[1,2†], Vanessa Mhanna[1,2†], Paul Stys[1‡], Helene Vantomme[1,2‡], Zhicheng Zhou[3‡], Federica Martina[2], Nicolas Coatnoan[2], Michele Barbie[2], Hang-Phuong Pham[4§], Béatrice Clémenceau[5], Henri Vie[5], Mikhail Shugay[6], Adrien Six[1], Barbara Brandao[3], Roberto Mallone[3,7], Encarnita Mariotti-Ferrandiz[1], David Klatzmann[1,2*]

[1]Sorbonne Université, INSERM, Immunology-Immunopathology-Immunotherapy, Paris, France; [2]AP-HP, Hôpital Pitié-Salpêtrière, Clinical Investigation Center for Biotherapies (CIC-BTi) and Immunology-Inflammation-Infectiology and Dermatology Department (3iD), Paris, France; [3]Université Paris Cité, Institut Cochin, CNRS, INSERM, Paris, France; [4]ILTOO pharma, Statistical department, Paris, France; [5]CRCINA, INSERM, CNRS, Université d'Angers, Université de Nantes, Nantes, France; [6]Center of Life Sciences, Skoltech, Moscow, Russian Federation; [7]Assistance Publique Hôpitaux de Paris, Service de Diabétologie et Immunologie Clinique, Cochin Hospital, Paris, France

*For correspondence: david.klatzmann@sorbonne-universite.fr

†These authors contributed equally to this work
‡These authors also contributed equally to this work

Present address: §Parean biotechnologies, Saint-Malo, France

**Abstract** T-cell receptors (TCRs) are formed by stochastic gene rearrangements, theoretically generating >$10^{19}$ sequences. They are selected during thymopoiesis, which releases a repertoire of about $10^8$ unique TCRs per individual. How evolution shaped a process that produces TCRs that can effectively handle a countless and evolving set of infectious agents is a central question of immunology. The paradigm is that a diverse enough repertoire of TCRs should always provide a proper, though rare, specificity for any given need. Expansion of such rare T cells would provide enough fighters for an effective immune response and enough antigen-experienced cells for memory. We show here that human thymopoiesis releases a large population of clustered CD8+ T cells harboring α/β paired TCRs that (i) have high generation probabilities and (ii) a preferential usage of some V and J genes, (iii) which CDR3 are shared between individuals, and (iv) can each bind and be activated by multiple unrelated viral peptides, notably from EBV, CMV, and influenza. These polyspecific T cells may represent a first line of defense that is mobilized in response to infections before a more specific response subsequently ensures viral elimination. Our results support an evolutionary selection of polyspecific α/β TCRs for broad antiviral responses and heterologous immunity.

## Editor's evaluation

This manuscript reports novel and valuable observations regarding human CD8 T cells that express shared T cell receptors amongst individuals and exhibit poly-specificity directed mainly to several unrelated viral antigens. In the revised version, the authors have addressed the majority of objection raised by the reviewers and the additional results and revised discussion provide a solid support to the authors claims. The results from these studies will add to the ongoing debate on T cell specificity by providing material for an informed decision on whether the data represent genuine cross-reactivity or technical confounders.

## Introduction

Specificity is considered a hallmark of the adaptive immune response. For T cells, specificity is mediated by T-cell receptors (TCRs), which interact with peptides presented by major histocompatibility complex (MHC) molecules through their complementary-determining-region-3 (CDR3). TCRs are formed by rearrangements between hundreds of gene segments at the alpha and beta loci, theoretically generating >$10^{60}$ possible sequences (*Dupic et al., 2018*; *Murphy and Weaver, 2016*; *Murugan et al., 2012*). Each rearrangement has its probability of generation, which has been modeled to vary by about 10 orders of magnitude for the beta chain alone (*Bradley and Thomas, 2019*). During their development within the thymus, T cells undergo a selection largely based on the strength of their activation by thymic antigen-presenting cells that eliminates >80% of them (*Klein et al., 2014*; *Sinclair et al., 2013*; *Vrisekoop et al., 2014*). For each individual, this process releases a pool of T cells expressing a repertoire of approximately $10^8$ unique TCRs (*Qi et al., 2014*).

How evolution shaped a process that selects cells that would efficiently respond to antigens that are not yet present, that is, from an infectious agent to come, is a central question of immunology. The paradigm is that the production and selection of a diverse enough repertoire of TCRs should always provide a proper, though rare, specificity for any given need. Expansion of such rare T cells would then provide enough fighters for an effective immune response and later enough antigen-experienced cells for immune memory (*Davis and Bjorkman, 1988*). Thus, it is widely accepted that '*T-cell-mediated immunity to infection is due to the proliferation and differentiation of rare clones in the preimmune repertoire that by chance express TCRs specific for peptide-MHC (pMHC) ligands derived from the microorganism*'(*Jenkins et al., 2010*).

However, this theory is challenged by overlooked data questioning the essence of specificity. First, a child with an X-linked severe combined immunodeficiency who had a reverse mutation in a single T-cell progenitor was protected from infections despite having a very restricted repertoire of only about 1000 different TCRs (*Bousso et al., 2000*). How such a repertoire ($10^{-5}$ of a normal repertoire size) could handle the numerous infections occurring during childhood awaits explanation. Second, robust experimental and epidemiological data have highlighted the importance of heterologous immunity, that is, immunity to one pathogen afforded by the exposure to unrelated pathogens (*Roth et al., 2005*; *Watkin et al., 2017*; *Welsh and Selin, 2002*). Likewise, for example, (i) memory T cells that are specific for one virus can become activated during infection with an unrelated virus (*Verhoeven et al., 2008*), (ii) there are numerous virus-specific memory phenotype T cells in unexposed individuals (*Su et al., 2013*), (iii) CMV infection enhances the immune response to influenza (*Furman et al., 2015*), and (iv) tumor-resident memory HBV-specific T-cell responses correlate with hepatocellular carcinoma outcome (*Cheng et al., 2021*).

We hypothesized that high-resolution sequencing of the TCR repertoire of developing thymocytes and peripheral T cells would yield insights into repertoire selection and function toward pathogens. We focused on the development of CD8+ T cells as they represent the main effectors against viral pathogens, those that have had most chances to influence the evolution of our immune system. We obtained samples from organ donors allowing access to thymocytes and peripheral T cells, as well as data sets from patients undergoing vaccinations or with infections; we also used TCR data available from multiple public repositories.

We identified a population of CD8+ T cells harboring diversified and polyspecific α/β TCRs that each can bind to multiple unrelated viral peptides, notably those from commonly encountered viruses such as EBV, CMV, or influenza. We hypothesize that these cells represent a first line of defense that is promptly mobilized in response to infections, possibly containing them before a more specific response subsequently ensures their control. Our results support an evolutionary selection of polyspecific α/β TCRs for broad antiviral responses and heterologous immunity.

## Results

### Thymopoiesis selects a large and diverse set of clustered CDR3s with high generation probabilities

To investigate how TCR diversity influences immune response to infectious agents, we started by analyzing the TCR repertoire dynamics of developing thymocytes. We first focused on the hypervariable CDR3 region of the TCR that interacts with the antigenic peptide, while CDR1 and CDR2 usually

interact with HLA molecules (*Rossjohn et al., 2015*). CDR3 analyses can therefore be used to investigate the sharing of TCR specificities across individuals with distinct HLA molecules. We analyzed the repertoire of purified CD4$^+$CD8$^+$CD3$^-$ (DPCD3$^-$), CD4$^+$CD8$^+$CD3$^+$ (DPCD3$^+$), and CD4$^-$CD8$^+$CD3$^+$ (CD8$^+$) thymocytes (*Figure 1—figure supplement 1A*). DPCD3$^-$ thymocytes represent the earliest stage of TCRβ-chain gene recombination, and their repertoire embodies the unaltered outcome of the TCR generation process; DPCD3$^+$ thymocytes are at an early stage of the selection process and their repertoire should be minimally modified; CD8$^+$ thymocytes have passed the selection process and bear a fully selected repertoire. We analyzed and represented the structure of these repertoires by connecting CDR3s (nodes) differing by at most one single amino acid (AA) (Levenshtein distance less than or equal to 1: LD≤1). LD≤1 connected CDR3s have been described to most often bind the same peptide (*Dash et al., 2017*; *Klinger et al., 2015*; *Madi et al., 2017*; *Chen et al., 2017*; *Glanville et al., 2017*; *Qi et al., 2016*; *Meysman et al., 2019*, *Figure 1—figure supplement 1B*). In such networks, connected CDR3s are designated as *clustered nodes* and the others as *dispersed nodes*. For normalization, we represented the first 18,000 most expressed β or α CDR3s from each sample.

We observed a significant increase in the number of clustered βCDR3s from DPCD3$^-$ to CD8$^+$ thymocytes (*Figure 1A and B*) (p<0.0001), which was remarkably consistent among all individuals studied, independently of their age, sex, or HLA (*Figure 1—figure supplement 2*). The degree of clustered CDR3s, that is, the number of neighbors based on LD≤1, was also significantly increased during T-cell differentiation for clustered βCDR3s (*Figure 1C and D*) (p<0.0001). Similar results were obtained for the αCDR3s (*Figure 1—figure supplement 3A–B*). These observations suggest a positive selection during thymopoiesis of TCRs with close CDR3 sequences, and therefore with shared recognition properties.

We evaluated the generation probability (Pgen) of each CDR3 pre- and post-selection (*Sethna et al., 2019*): (i) clustered TCRs from both DPCD3$^+$ and CD8$^+$ thymocytes have a significantly higher Pgen (p<0.0001) than the dispersed ones (*Figure 1E*); (ii) Pgen increased significantly (p<0.0001) from DPCD3$^+$ to CD8$^+$ thymocytes (*Figure 1E*), although the CDR3 length of DP and CD8$^+$ thymocytes showed mean distribution shifts of less than one AA difference (*Figure 1—figure supplement 3C*); (iii) there is a robust correlation between βCDR3 Pgen and their number of connections (p<0.0001) (*Figure 1F* and *Figure 1—figure supplement 3D*). However, it should be emphasized that the difference in the mean Pgen between clustered and dispersed TCRs is of about 20-fold, which is small when compared to the $10^{-\log10}$ span of the Pgen of the two populations. Likewise, many CDR3s with high Pgen do not cluster, indicating that a high Pgen is not the only nor most important driver of clustering (*Figure 1—figure supplement 3E and F*). Of note, we used the same model to evaluate Pgen of CDR3s from different individuals while it has been recently shown that the repertoire generation parameters for the immunoglobulin heavy chain differ between individuals (*Slabodkin et al., 2021*). However, such interindividual variation has not been reported for TCR repertoire generation.

Clustered βCDR3s belong to TCRs that have a preferential usage of V and J genes, with a significant (p<0.01) overrepresentation of Vβ 12-3, 27, 5-1, and 7-9 and of JB 1-1 and 2-7 (*Figure 1—figure supplement 3G and H*). This results in a clear separation of clustered versus dispersed TCRs in a principal component analysis (PCA) of VJ usages (*Figure 1G*).

## Clustered thymic CDR3s are enriched for publicness

Clustered βCDR3s are also enriched in public sequences, that is, being shared between at least two individuals (p<0.0001) (*Figure 2A*). The significant increase of public βCDR3s in CD8$^+$ versus DPCD3$^+$ thymocytes (p<0.0001) is mostly that of the clustered βCDR3s. For CD8$^+$ thymocytes, up to 31.7% of clustered βCDR3s are public compared to barely 1% of the dispersed ones (*Figure 2A*, *Figure 2—figure supplement 1A*). Public βCDR3 have a significantly higher Pgen than the private ones (p<0.0001) (*Figure 2B*), which is robust across all patients (*Figure 2—figure supplement 1B*, *Figure 2—source data 1*). Moreover, a high positive correlation was observed between Pgen and the number of connections, and between cross-individual sharing and the number of connections, and to a lesser extent between Pgen and cross-individual sharing (*Figure 2—figure supplement 1C–D*). Noteworthy, the number of public βCDR3 shared between individuals is independent of the number of shared HLA Class I alleles (*Figure 2C*). Finally, clustered βCDR3s of one individual can be connected to those of other individuals (up to 12), and more frequently in CD8$^+$ versus DPCD3$^+$ thymocytes (*Figure 2D and E*), indicating a convergence of specificities between individuals' clustered

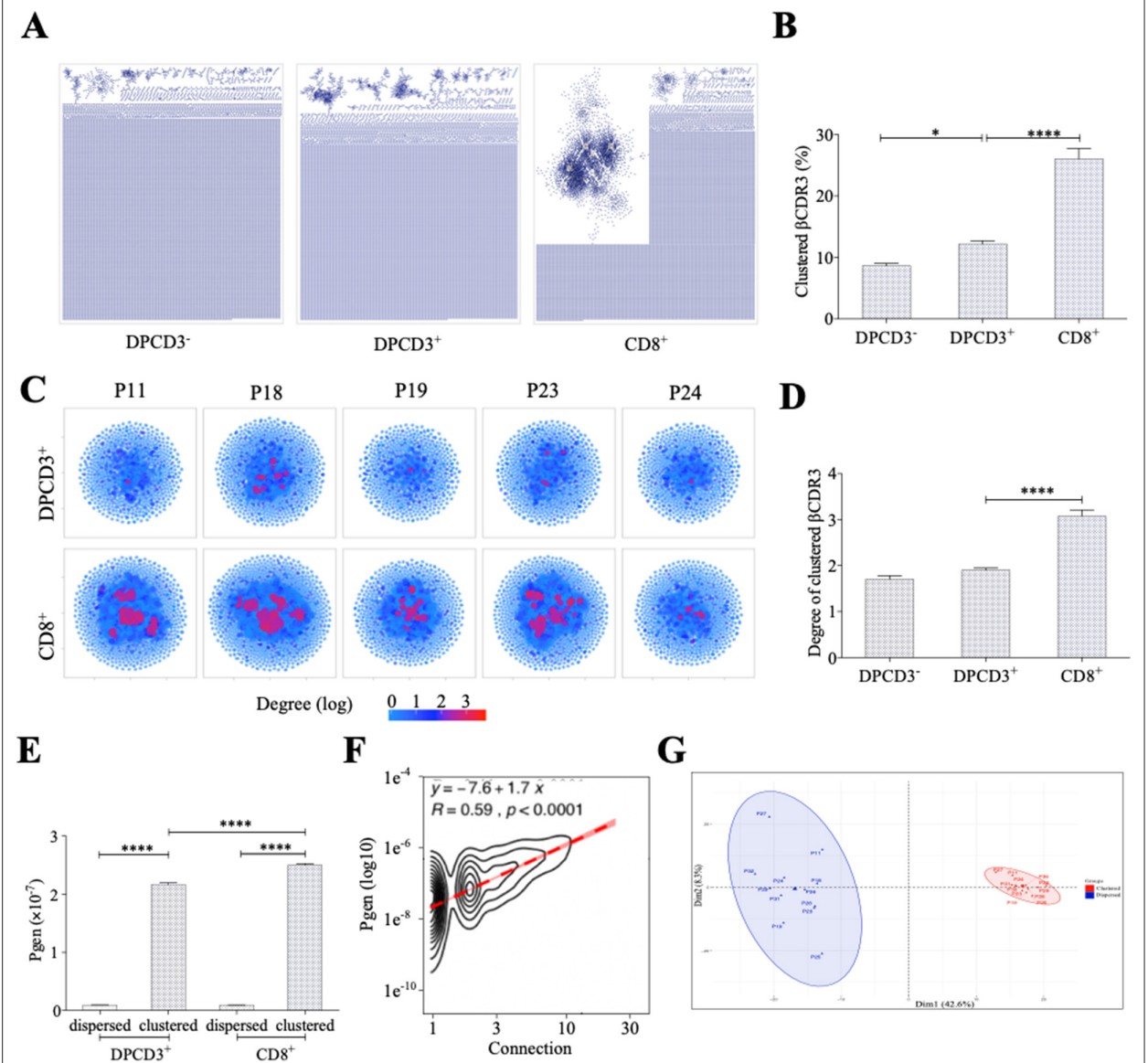

**Figure 1.** Thymocyte differentiation produces clustered CDR3s with high generation probability and preferential TRB VJ gene combinations.
(**A**) Representation of βCDR3aa networks from DPCD3⁻, DPCD3⁺, and CD8⁺. Each dot represents a single CDR3. Dots are connected (forming clusters) by edges defined by Levenshtein distance of ≤1 (one AA substitution/insertion/deletion). (**B**) Percentage of clustered βCDR3aa (*p=0.0152 and ****p<0.0001, Mann–Whitney test, mean ± s.e.m, n=13). (**C**) βCDR3aa clustered from DPCD3⁺ and ThyCD8. Each dot represents a single βCDR3aa. The colour scale represents the number of neighbors for each CDR3. Blue dots have only one connection while red dots have more than three connections (up to 30). (**D**) Degree of clustered βCDR3aa (****p<0.0001, Mann–Whitney test, mean ± s.e.m, n=13). (**E**) Generation probability of dispersed and clustered βCDR3aa in DPCD3⁺ or CD8⁺ cells (****p<0.0001, Mann–Whitney test, mean ± s.e.m, n=13). (**F**) Correlation between *Pgen* and βCDR3 number of connections in the CD8⁺ thymocyte repertoire. Contour plot represent the generation probability as a function of βCDR3 connections in the CD8⁺ thymocytes for donor P29. Linear regression curves between *Pgen* and number of connections are represented as red dotted lines ('y' represent the regression curve's equation). Pearson correlation coefficient 'R' and p value 'p' are calculated for each individual (**Figure 3D**). (**G**) PCA analysis of TRB VJ gene combinations in CD8 thymocytes. Blue: dispersed nodes; Red: clustered nodes. AA, amino acid; PCA, principal component analysis.

The online version of this article includes the following figure supplement(s) for figure 1:

**Figure supplement 1.** Dotplot of thymocytes sorting.

**Figure supplement 2.** Properties of thymocytes' TCRs.

**Figure supplement 3.** Properties of thymocytes' TCRs.

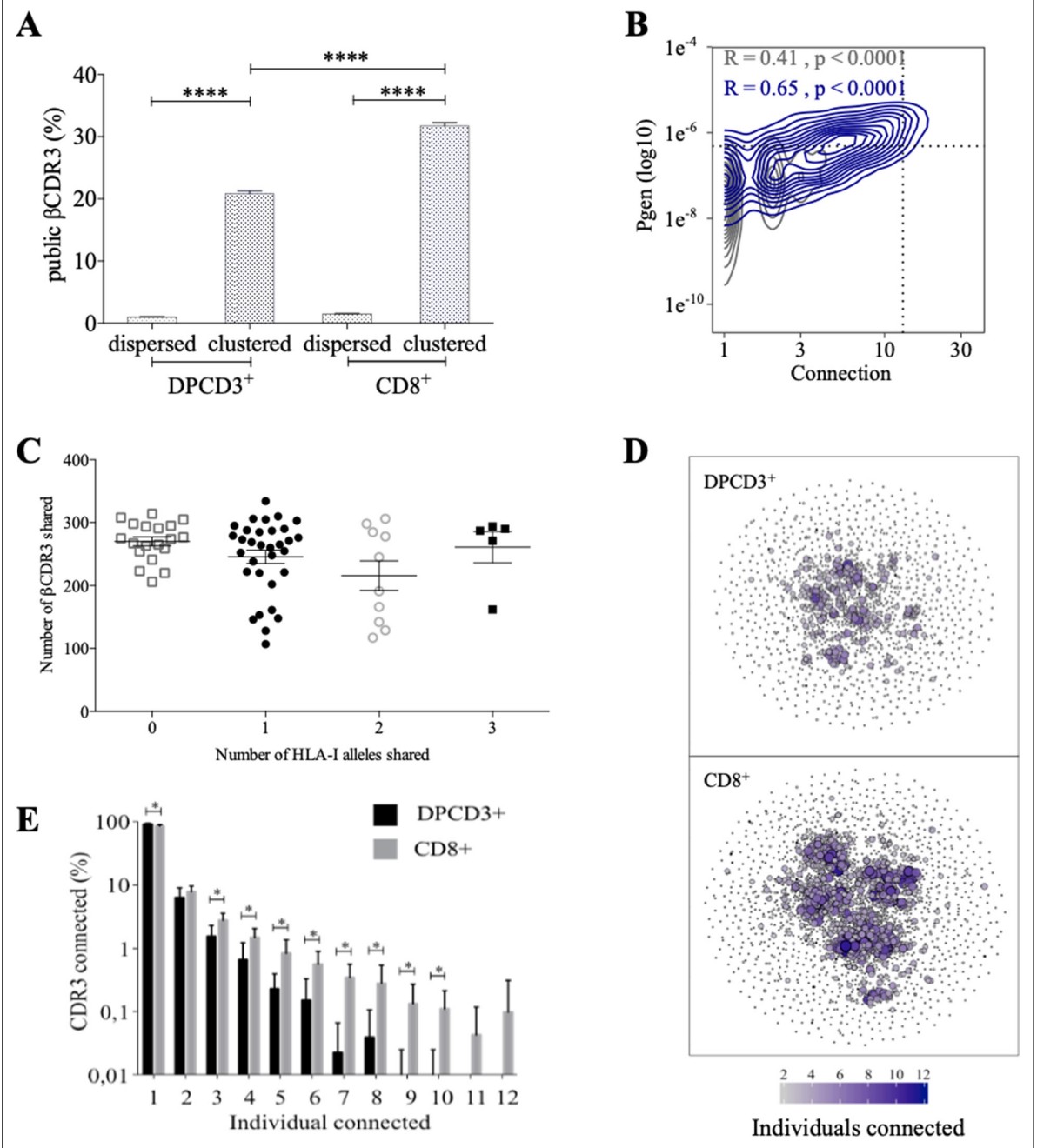

**Figure 2.** Thymocyte differentiation produces clustered CDR3s with high publicness. Mean percentages of public (black) or private (gray) βCDR3s in all, dispersed or clustered nodes (****p<0.0001, Mann–Whitney test, mean ± s.e.m, n=13). (**B**) Enrichment of public βCDR3s in the CD8[+] thymocyte repertoires. Representation of the generation probability as a function of βCDR3 connections in individuals (Pn). The contour plots represent shared (blue) or private (gray) βCDR3s for a representative patient. The Pearson correlation coefficient 'R' and p value 'p' are calculated for each group. The black dotted lines delimit the threshold for the 2.5% sequences with the higher *Pgen* and connection. βCDR3s with both the highest *Pgen* and connections are also the most public for 12 out of 12 individuals (p<0.0001, Chi-square test, *Figure 2—source data 1*, *Figure 2—figure supplement 1B*). (**C**) The number of public βCDR3s in CD8[+] thymocytes is independent of the number of HLA-I alleles shared. Each dot represents the number of βCDR3s shared between two donors in the first 18,000 CD8[+] thymocytes. There is no significant difference in the number of public βCDR3s according to the number of HLA-I alleles shared (p>0.1, Mann–Whitney test, mean ± s.e.m, n=13). The number of public βCDR3s is independent of the number of HLA alleles shared. (**D**) Public convergence of βCDR3 recognition properties during thymopoiesis. CDR3 connections between individuals. The top 1500 βCDR3s were sampled from DPCD3[+] (upper square) and CD8[+] (lower square) cells from each individual and pooled. The CDR3s are clustered based on LD≤1 with colour and size both representing the level of sharing between individuals for each CDR3. (**E**) Convergence of public βCDR3 specificities during thymopoiesis. Bar plots representing the percentage of CDR3s from an individual that are connected to CDR3s of other individuals, for DPCD3[+]

*Figure 2 continued on next page*

*Figure 2 continued*

and CD8[+] thymocytes. The first two bars represent CDR3s that are not connected (n=1). The number of unconnected nodes in DPCD3[+] is higher than in CD8[+] (*p=0.002). The other bars represent the percentage of CDR3s connected between individuals. The number of nodes connected to 3–10 individuals is significantly higher in CD8[+] than in DPCD3[+] cells (*p<0.01, multiple t test).

The online version of this article includes the following source data and figure supplement(s) for figure 2:

**Source data 1.** Enrichment of public βCDR3s in CD8[+] thymocytes versus DPCD3[+].

**Figure supplement 1.** Publicness of thymocytes' CDR3s.

repertoires. Altogether, these results indicate that the mechanisms for TCR generation and for their further thymic selection are biased to shape a public repertoire of connected βCDR3s with shared recognition properties.

## Clustered public thymic CDR3s are enriched in public viral specificities

The preferential selection of clustered public CDR3s that could represent over 8% of the sampled repertoire (*Figure 2A*) raises the question of their specificities. As the main function of CD8[+] T cells is cytotoxicity toward virally infected cells, we investigated whether the clustered CDR3s could be associated with virus recognition. We curated two databases of βCDR3s from TCRs specific for human infectious pathogens (*Tickotsky et al., 2017*; *Shugay et al., 2018*) to retain only a set of 5437 βCDR3 that had been identified by the binding of soluble multimeric MHC/peptide complexes, that is, tetramers or dextramers (*Dolton et al., 2014*). We detected an enrichment of these virus-specific βCDR3s in clustered versus dispersed CD8[+] thymocytes (p<0.0001) and in CD8[+] versus DPCD3[+] thymocytes (p<0.0001) (*Figure 3A*). Moreover, these virus-specific βCDR3s were significantly enriched in βCDR3s with the highest *Pgen* and number of connections (*Figure 3B and C*, *Figure 3—figure supplement 1A*, *Figure 3—source data 1*). They were also highly shared between individuals (*Figure 3D*) and they contained a high representation of CMV, EBV, and influenza specificities (*Figure 3—figure supplement 1B*). To better represent the contribution of virus-specific CDR3s to the global repertoire of CD8[+] cells, we connected them to all CD8[+] cell CDR3s of the 12 individuals using either a perfect matching (LD=0) (*Figure 3E*, left panel) or an LD≤1 (*Figure 3E*, right panel). In the latter representation, the 13,557 connected CDR3s (i) covered most of the clustered CDR3s of all individuals (*Figure 3E*, *Figure 3—figure supplement 2A*), (ii) amounted to up to 7.4% and 12.3% of the DPCD3[+] and CD8[+] cells CDR3 repertoires, respectively (*Figure 3F*), and (iii) were highly shared between individuals (*Figure 3—figure supplement 2B*).

Altogether, these results indicate that the selection of clustered CDR3s with high generation probabilities and high inter-individual sharing corresponds, at least in part, to the selection of virus-specific TCRs whose CDR3s are remarkably conserved between individuals independently of their HLA restriction and form what could be referred to as a 'paratopic network'.

## Identification of thymic polyspecific TCRs

After thymic selection, an in-silico estimation indicated over 10% of the CD8[+] thymocytes' TCR repertoire as specific for the small set of 30 tested peptides derived from seven viruses that was used for the analysis (*Figure 3F*). This appears hardly compatible with the fact that the TCR repertoire should contain TCRs that react specifically to countless antigens. Actually, we observed that a single cluster of βCDR3s from CD8 thymocytes could comprise unique βCDR3s from TCRs that had been assigned to different viral specificities in independent experiments, for example, to CMV, influenza, EBV, and yellow fever virus (YFV). This observation is puzzling, as clustered CDR3s are likely to respond to similar antigens (*Figure 3—figure supplement 1B*, *Dash et al., 2017*; *Klinger et al., 2015*; *Madi et al., 2017*; *Chen et al., 2017*; *Glanville et al., 2017*; *Qi et al., 2016*; *Meysman et al., 2019*). It suggested that some virus-associated TCRs could have a 'fuzzy' specificity that could allow the recognition of peptides from different viruses. We explored the public databases of virus-specific TCRs (*Tickotsky et al., 2017*; *Shugay et al., 2018*) in more detail and discovered that numerous single CDR3s had been assigned to multiple specificities, for example, single CDR3s binding both CMV and influenza tetramers (*Figure 3—figure supplement 3A*). In addition, among the 13,557 virus-associated βCDR3s that we previously estimated in CD8[+] thymocytes (*Figure 3E*, 'LD≤1'), almost 7% (958/13557) are in silico assigned (LD≤1) as specific for at least two distinct viruses (*Figure 3—figure supplement 3B*).

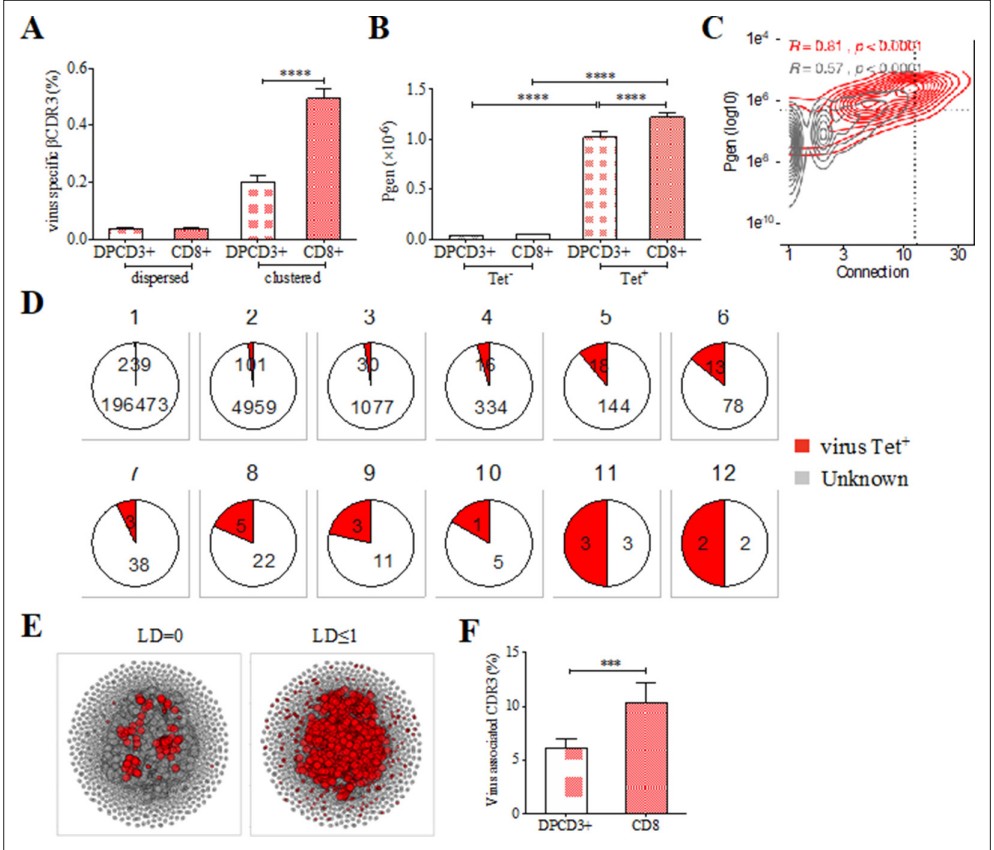

**Figure 3.** Clustered public TCRs are enriched for virus-specific TCRs. (**A**) Barplots showing the mean percentages of CDR3s associated with pathogens within DPCD3+ and ThyCD8 cells and for dispersed or clustered CDR3s. (**B**) Mean generation probability of virus-specific βCDR3s, based on their identification by tetramer, in DPCD3+ or CD8+ thymocytes (A,B, ****p<0.0001, Mann–Whitney test, mean ± s.e.m, n=13). (**C**) Enrichment of virus-specific βCDR3s in the CD8+ thymocyte repertoire. Representation of the generation probability as a function of βCDR3 connections in a representative individual. The contour plot represents βCDR3s from TCRs identified as virus-specific based on tetramer identification *Roth et al., 2005*, *Watkin et al., 2017* (red) or with unknown specificity (gray). Pearson correlation coefficient 'R' and p value 'p' are calculated for each group. The black dashed lines delimit the threshold for the 2.5% sequences with both higher *Pgen* and degree of connection. βCDR3s with both the highest *Pgen* and connections were also the most virus-specific for 11 out of 12 individuals, *Figure 3—figure supplement 1A* (p value<0.0001, Chi-square test). (**D**) Virus-specific βCDR3s sharing in CD8+ thymocytes. Pie charts represent the βCDR3s from private (*Dupic et al., 2018*) to shared βCDR3s by all donors (*Roth et al., 2005*), in gray for βCDR3s with unknown specificity or in red for those with a virus specificity. (**E**) Network of virus-associated CDR3s. The shared (identical) and linked (LD≤1) virus-associated CDR3s within the CDR3 network of one individual are in red. (**F**) Virus antigen coverage. Barplot represents the mean percentage of LD≤1 to virus-associated CDR3s in all individuals (***p<0.001, Mann–Whitney test, mean ± s.e.m, n=13). TCR, T-cell receptor.

The online version of this article includes the following source data and figure supplement(s) for figure 3:

**Source data 1.** Enrichment of virus-specific βCDR3s from databases in clustered CD8+ thymocytes.

**Figure supplement 1.** Virus specificities of thymocytes' CDR3s.

**Figure supplement 2.** Sharing of virus specific thymocytes' CDR3s.

**Figure supplement 3.** Binding properties of single-cell TCRs.

## Binding properties of thymic polyspecific TCRs

We further investigated the T-cell specificities of CD8+ thymocytes by using the GLIPH2 algorithm to analyze a combined data set of (i) 216,000 CDR3β sequences from our CD8+ thymocytes and (ii) 32,496 tetramer-specific sequences from the curated VDJdb database (*Shugay et al., 2018*). The GLIPH2 algorithm groups together CDR3s that have structural characteristics that make them likely to recognize the same antigens, hence called 'specificity groups' (*Glanville et al., 2017*). GLIPH2

identified 93,182 such specificity groups. Noteworthy, 31.6% of these specificity groups contained at least one CDR3 with a known viral specificity. Even more surprisingly, 6.8% of GLIPH2 specificity groups contained CDR3s with more than one specificity, and up to nine different ones (*Figure 4A*). Thus, GLIPH2 identified among CD8⁺ thymocytes' TCRs a large number of clusters with viral specificities, among which a significant fraction appeared polyspecific.

## Identification and binding properties of paired α/β polyspecific TCRs from single peripheral cells

As these observations are made from studying unpaired CDR3s, we aimed to confirm polyspecificity with paired α and β virus-specific CDR3s obtained from single-cell TCR sequencing. These sequences were obtained from 160,914 blood CD8⁺ T cells isolated from four healthy donors and incubated simultaneously with dextramers complexed with peptides from CMV, EBV, HIV, HPV, HTLV, and influenza (see Materials and methods).

We could identify numerous single cells harboring TCRs that did bind to multiple dextramers, which others also observed but excluded from their analysis (*Zhang et al., 2021*). For example, single-cell derived TCRs that bind HLA-matched dextramers loaded with EBV or CMV peptides are shown (*Figure 4B*). Projecting the binding of EBV dextramers onto a t-SNE representation based on single-cell specificity identified a few distinct regions that contained CMV (blue) and EBV positive cells (red), and also cells bound to both CMV and EBV dextramers (violet). As another example, a chord diagram representation of the binding of unique TCRs to HLA-matched EBV dextramers loaded with different EBV peptides (*Figure 4C*, in green) shows that a large proportion of the TCRs that bind to the RAKFKQLL peptide also bind the other EBV peptides. Furthermore, some TCRs recognizing this RAKFKQLL peptide can also bind peptides from CMV (red), HIV (dark blue), or Flu (dark gray) (*Figure 4C*). Altogether, these analyses reveal unique cells that bind to dextramers presenting different unrelated peptides.

To better understand this phenomenon, we analyzed in more detail the dextramer binding scores for 2074 cells from two individuals that all use the same CASSIRSSYEQYF βCDR3 associated with different αCDR3s, altogether representing 131 unique TCRs. We analyzed the binding score of these cells for all the 49 different dextramers (*Figure 4—figure supplement 1A*), of which only those with a relevant binding are represented in *Figure 4D*. Almost all of these 2074 cells had the highest binding score for a matched HLA-A*0201 dextramer bound to an influenza peptide, indicating that the strong binding of this dextramer was mostly driven by this βCDR3. However, approximately half of these 2074 cells also had a noticeable but weaker binding score for the three HLA-A*0301 and two A*1101 dextramers presenting peptides from either CMV or EBV. Importantly, each of these five dextramers had their own unique binding patterns, resulting in TCRs that could bind only one of these dextramers to TCRs binding them all or none. Also, these same dextramers do not have the same binding patterns for other TCRs using different CDR3s. Likewise, for example, we identified a CASS-LYSATGELFF βCDR3 that has a strong binding for only one of these five dextramers when associated with many different αCDR3s and a weaker binding to the four others (*Figure 4—figure supplement 1B*). We also identified one βCDR3 that binds to four out of five of these dextramers only when associated to a specific αCDR3 (*Figure 4—figure supplement 1C*). Thus, these peculiar binding patterns do not appear to reveal a nonspecific background staining, but rather indicate a TCR dependency of these weaker dextramer bindings.

We analyzed dextramer bindings in more details for the most represented clonotypes in donor 2 (*Figure 4E*). This clearly identifies 'strong' and 'weak' binders, with some heterogeneity of binding for the different cells from a given clonotype. These weak bindings appear HLA-dependent as, for example, they are mostly seen for A03 and A011 dextramers, and are almost nonexistent for A02 or B08 dextramers. They also appear peptide-dependent as for the same HLA, the binding patterns are different according to the peptide presented. In addition, we observed that the binding intensity for weak binders is statistically lower when the cells also bind a strong binder (*Figure 4—figure supplement 2A*), indicating that the weak bindings are competed for by strong ones. This is important as in this experiment the cells were co-incubated with a mixture of all dextramers that all competed for available TCRs. Altogether, the peptide and HLA-dependent nature of the weak bindings and their competition by strong binders indicate that they are TCR mediated.

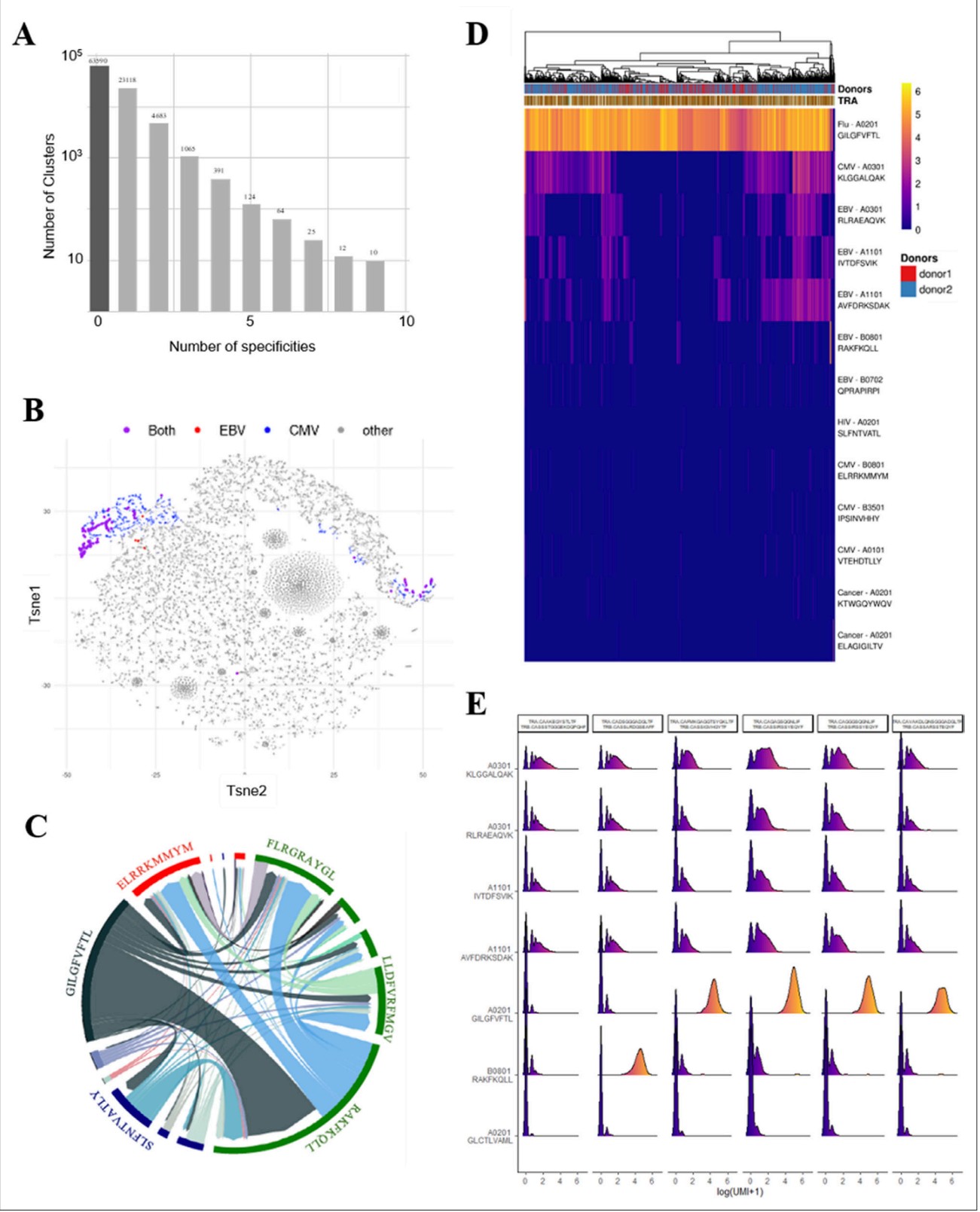

**Figure 4.** Identification of polyspecific TCRs. (**A**) Analysis of shared T-cell specificities with the GLIPH2 algorithm. 216,000 CDR3β sequences from the CD8 thymocytes and 32,496 sequences from VDJdb that are tetramer specific were analysed to obtain 93,182 specificity groups. 31.6% of the groups contained at least one CDR3 with a known specificity for a virus. 6.8% of the clusters identified by GLIPH2 contained CDR3s with more than one specificity, and up to nine different ones. (**B**) t-SNE plots of βCDR3 sequences from peripheral CD8 T cells based on sequence similarity. On this

*Figure 4 continued on next page*

*Figure 4 continued*

overall repertoire representation, we labeled cells that were annotated as EBV specific (red), CMV specific (blue), and ones that bound to both EBV and CMV dextramers (purple). (**C**) Chord diagram showing TCR binding to multiple Dextramers. The chord diagram shows TCR binding to HLA-matched dextramers loaded with peptides from unrelated viruses. Each segment represents TCR binding to the peptides marked above. The size of the segments corresponds to the number of TCRs binding to these peptides. The link between segments identifies multiple TCR binding to different peptides. The colours of the segments represent the different viruses: CMV (red), EBV (green), HIV (dark blue), HPV (light green), HTLV (purple), and influenza (dark gray). The full list of the different peptides is in *Figure 4—source data 1*. (**D**) Heatmap of the binding score (in a logarithmic scale) for different dextramers of different TCRs using the same CASSIRSSYEQYF βCDR3, in two donors. The polyspecificity is oriented toward the recognition of common viruses such as influenza, EBV, and CMV. Density distributions of the binding score (in a logarithmic scale) for different dextramers of the top clonotypes in donor 2. TCR, T-cell receptor.

The online version of this article includes the following source data and figure supplement(s) for figure 4:

**Source data 1.** List of peptides represented on the chord plot from *Figure 4C*.

**Figure supplement 1.** Binding patterns to A*03 and A*11 dextramers.

**Figure supplement 2.** Single-cell gene expression analysis.

We could not relate these weak binding patterns to other surface proteins that can interact with MHC, such as KIR molecules. This was directly confirmed by analyzing KIR expression in the transcriptome of these cells. There was much fewer KIR$^+$ cells than there were polyspecific ones (*Figure 4—figure supplement 2B and C*) and there was no association between KIR expression and polyspecificity. Additionally, we showed that these weak binding patterns could not be linked to the CD45 RA naïve/quiescent versus CD45 RO memory/activated phenotype of the cells, nor to any other relevant genes (*Figure 4—figure supplement 2D and E*). Altogether, these results indicate that beside a 'prime' dextramer binding, presumably identifying their main specificity, some 'polyspecific' TCRs can bind weakly to additional dextramers.

## Polyspecific T cells are activated in vitro by multiple viral peptides

These peculiar properties of polyspecific TCRs led us to assess their functional relevance. We first used human effector memory CD8$^+$ T cells that were purified according to their binding of CMV or EBV HLA-matched dextramers (*Figure 5—figure supplement 1*); the sorted cells were then stimulated by either the peptide that was used to purify them, or by different ones, all at the physiologically relevant 1 µM concentration, and their activation was measured by their IFN-γ production (*Figure 5A*). All T cells were efficiently nonspecifically activated by PMA/ionomycin. Control T cells that did not bind CMV nor EBV dextramers could not be stimulated by related peptides. In contrast, dextramer-sorted cells could be activated by their cognate peptide and almost as well if not better by the other (*Figure 5B*).

We also tested the response of bulk T cells developed as cell therapy products for the treatment of CMV infections in humans. These cells had been selected through 18 days of stimulation in the presence of a peptide pool covering the complete sequence of the pp65 protein of CMV. Upon stimulation by their cognate peptides, over 85% of the CD8$^+$ T cells produced IFNγ (*Figure 5—figure supplement 2A*). Surprisingly, a small fraction of the same cells produced interferon upon stimulation with unrelated CMV-IE1 peptides, and a large fraction upon stimulation with a peptide pool which covers the sequence of the BZLF-1 protein of EBV (*Figure 5—figure supplement 2A*). There was no interferon production upon stimulation of the same cells by a peptide pool which covers the EBNA1 protein of EBV (*Figure 5—figure supplement 2A*). The stimulation with EBV peptides requires a 100-fold higher but still physiological concentration of peptides (*Figure 5—figure supplement 2B*).

Finally, we directly confirmed the polyspecificity of unique TCRs by their cloning and re-expression. We used a 5KC murine T-hybridoma cell line devoid of endogenous TCR, and transduced with a ZsGreen fluorescent reporter under the control of nuclear factor of activated T cells (NFAT) and with human CD8 (*Mann et al., 2020*). These 5KC transductants were then stimulated with K562 cells transduced with either HLA-A*02:01 or HLA-A*03:01 as antigen-presenting cells, in the presence of individual peptides at increasing concentrations. From the single-cell sequencing data set, we selected two TCRs with similar binding patterns for dextramers loaded with Flu, CMV and EBV peptides, and which differ by only two AAs in the CDR3beta region while using different alpha chains. TCR#35–13 responded strongly to HLA-A*02:01 loaded with Flu MP 58–66 (EC50: 2.02×10$^{-9}$ M) and CMV IE1-184-192 (EC50: 7.99×10$^{-7}$ M), and poorly to EBV BMLF1 280–288 (EC50: 9.37×10$^{-3}$ M) (*Figure 5C*,

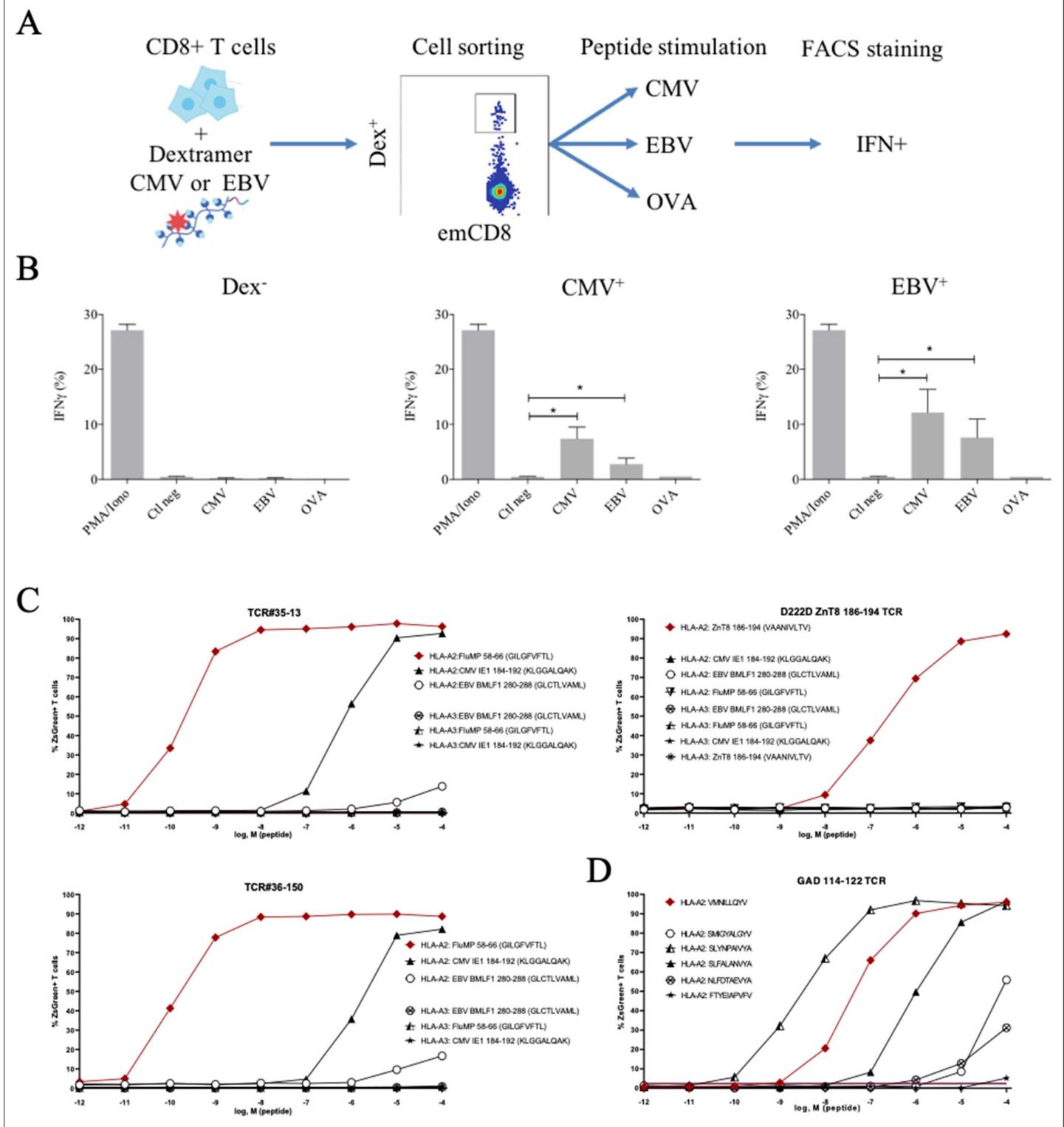

**Figure 5.** Polyreactivity of polyspecific TCRs. (**A**) Schematic representation of the in vitro cross-activation experiment. (**B**) In vitro activation of polyspecific T cells. Percentage of IFNγ producing emCD8⁺ cells after activation with PMA/ionomycin (positive control), no peptide (Ctl neg) or CMV, EBV, and OVA peptides (*p<0.05, Mann–Whitney test, mean ± s.e.m.). (**C**) Poly-reactivity of re-expressed viral epitope-reactive TCRs. 5KC cells transduced with TCR#35–13 (upper left), TCR#36–150 (bottom left) that responded to different peptides in the single-cell screening, and D222D-ZnT8186–194 (upper right) that responded to its cognate peptide ZnT8186–194 were stimulated with the indicated nonamer peptides at different concentrations, pulsed on K562 cells expressing either HLA-A*02:01 or HLA-A*03:01. The percent cells expressing the NFAT-driven ZsGreen reporter

*Figure 5 continued on next page*

*Figure 5 continued*

is shown as activation readout. A representative experiment out of two performed is shown.(**D**) TCRs from single-sorted CD8⁺ T cells stained with tetramers loaded with pancreatic self-peptides were cloned in reporter cell lines. We analyzed their response to various peptides: cognate peptide in red and peptides with no significant structural commonalities (gray). TCR, T-cell receptor.

The online version of this article includes the following source data and figure supplement(s) for figure 5:

**Source data 1.** List of the sequences of individual TCRs expressed and analyzed in vitro in *Figure 5*.

**Figure supplement 1.** Gating strategy for sorting effector memory (emCD8) Dextramer positive cells.

**Figure supplement 2.** Polyspecific properties of anti-CMV cytotoxic cells.

upper left). TCR#36–150 had very similar responses for HLA-A*02:01 loaded with Flu MP 58–66 (EC50: $9.99\times10^{-11}$ M), CMV IE1-184-192 (EC50: $7.99\times10^{-7}$ M), and EBV BMLF1 280–288 (EC50: $9.37\times10^{-3}$ M) (*Figure 5C*, bottom left). None of the TCRs responded to peptide-pulsed K562 expressing HLA-A*03:01. In contrast, a control D222D TCR (*Culina et al., 2018*) known as specific for a ZnT8 186–194 peptide responded to this peptide (EC50: $2.68\times10^{-7}$ M), but not to the Flu, EBV, or CMV peptides (*Figure 5C*, upper right).

We then further analyzed TCRs derived from single-sorted CD8⁺ T cells stained with tetramers loaded with pancreatic self-peptides. TCRs with known specificities were also cloned and expressed in the reporter cell line. We analyzed their response to a set of peptides comprising their cognate peptide and peptides with no significant structural commonalities, selected by testing combinatorial peptide libraries (*Mann et al., 2020*). We also observed a marked polyspecificity of these TCRs. For example, a TCR identified as responding to the VMNILLQYV peptide from the Glutamic Acid Decarboxylase protein responded even better to the structurally unrelated SLYNPAIVYA peptide from human chondroitin sulfate N-acetylgalactosaminyltransferase 2 (*Figure 5D*).

## Discussion

Our work brings to attention an overlooked phenomenon hidden in the literature. In a simple curation of public data sets of antiviral TCRs that were assigned to respond to known viruses—retaining only those from cells that could bind or respond to a known peptide—we found a surprisingly frequent representation of CDR3s that had been assigned to different viral specificities from independent experiments. Within the 40,939 curated βCDR3, 24% were from TCRs that could bind at least two totally unrelated viral peptides. This observation is highly relevant, as it arises in part from data sets of full TCRs from single-cell sequencing. Noteworthy, in these latter experiments, cells are co-incubated with multiple dextramers bearing a large set of unrelated peptides; although this data set has been widely analyzed, researchers have missed (or dismissed) that a single cell could bind multiple dextramers (sometimes a dozen) possibly because some of these bindings are weak. Actually, in the methodology description, the technology provider recommends determination of the specificity of a given T cell by the identity of the dextramer that has the highest binding counts on that cell. This is a rather crude readout, as the highest count could be marginally so. Furthermore, as there are limited numbers of HLA molecules available per cell, there is competition between dextramers for binding at the cell surface during their co-incubation. We actually observed such competition in which dextramers with weak bindings are competed for by those with strong ones (*Figure 4—figure supplement 2A*).

That a single cell binds multiple dextramers bearing unrelated peptides is in our view meaningful. As the binding of multimeric HLA/peptide complexes is a paradigmatic method to define specificity, multiple binding indicates multiple specificities. We thus defined TCRs binding multiple HLA/peptide complexes as polyspecific, and by extension called the cells harboring these TCRs as polyspecific T cells (psT cells). The polyspecific nature of these TCRs translates in polyreactivity as we show here that cloned polyspecific TCRs indeed trigger cell activation in response to multiple peptides, although with different efficiencies and with not always a correlation between the dextramer binding and its reactivity. This is likely due to the fact that cell activation assays have better sensitivity and robustness than dextramer binding assays.

In this line, two cloned TCRs with beta CD3s differing by 2AA and associated with different alpha CDR3s have a remarkably quantitatively similar activation profile for three different unrelated peptides (*Figure 4E*), highlighting the assay robustness. Incidentally, it also emphasizes the validity of the

Levenshtein distance parameter as a proxy of TCRs with similar antigen recognition properties. In contrast, it has been reported that some peptides that could not properly bind HLA were nevertheless capable of eliciting efficient CD8[+] effector T-cell responses capable of eradicating tumors expressing the cognate antigen (*Ebrahimi-Nik et al., 2021*). Also, we repeatedly detected a strong, dose-dependent and saturable response to an A02-KLGGALQAK dextramer which was used as a 'negative control' for A03-KLGGALQAK. This is surprising as this peptide is predicted to be a very poor binder for HLA-A2. There are however several examples of peptides with these low predicted HLA binding scores that nonetheless show some experimental binding to HLA, sufficient for refolding recombinant HLA molecules (*Gonzalez-Duque et al., 2018*; *Azoury et al., 2020*; *Azoury et al., 2021*). The response to the A02 or A03 presented KLGGALQAK peptide illustrates how HLA binding affinity does not always predict the TCR-mediated activation. Moreover, it is known that the TCR avidity threshold for functional responses is notoriously lower than that required for dextramer binding (*Rius et al., 2018*). Altogether, this indicates that T-cell activation results are more relevant for discussing TCR reactivity.

A limitation of our work is that functional studies could not be performed with thymocytes' TCRs as these were obtained by bulk sequencing and thus not as paired αβTCRs. Nonetheless, the polyspecific/polyreactive TCRs from the single-cell data set necessarily originated from a successful thymocyte differentiation and thus the observation made with CD8-derived peripheral T cells TCRs should apply to thymocytes derived TCRs. Also, the polyspecificity of TCRs translate in a polyreactivity of the T cells expressing them, although at peptide concentrations that can span >6 log10 but that are considered physiological (*Rius et al., 2018*). It thus remains to investigate how these different sensitivity of response impact polyreactivity at work.

TCRs are notoriously 'cross-reactive', and it is estimated that each TCR could potentially recognize >10[6] different peptide/MHC combinations (*Adams et al., 2016*; *Birnbaum et al., 2014*; *Mason, 1998*; *Wooldridge et al., 2012*). Indeed, the HLA/peptide/TCR *ménage à trois* leaves room for mimotopic cross-reactivity, that is, the fact that two unrelated peptides can bind differently to a given HLA molecule, but altogether generate a globally similar structure that could be recognized by the same TCR. Also, it was recently shown that peptides need to share only five residues at specific positions to bind the same TCR (*Wooldridge et al., 2012*; *Nelson et al., 2015*). This latter observation has functional relevance as bacterial-derived peptides sharing such five residues with a tissue-restricted MOG self-antigen were shown (i) to activate a known MOG-specific TCR as well as a MOG peptide and (ii) to trigger an autoimmune attack against brain tissues upon mouse immunization (*Nelson et al., 2015*; *Wucherpfennig and Strominger, 1995*). Thus, a large part of cross-reactivity is explained by a structural conservation of the TCR interaction surface by different peptides (*Birnbaum et al., 2014*). The functional relevance of these observations was described as that '*TCR cross-reactivity enables effective surveillance of diverse self and foreign antigens without necessitating degenerate recognition of non-homologous peptides*' (*Birnbaum et al., 2014*).

In contrast to this mimotopic cross-reactivity, we observed polyspecificity using a very restricted pool of peptides with no specific AA sharing conservation. According to Don Mason's estimates, each TCR may respond to over 10[6] different peptides from an estimated repertoire of >10[10] possible peptides, making the chance to find a cross-reactive peptide at random around 10[−4]. Here, the chance to find even just one cross-reactive peptide among these few peptides that we tested would be around 10[−3], the chance to find two cross-reactive peptides around 10[−6] and to find three or more cross-reactive peptides infinitesimal. Thus, if the polyreactivity that we described is part of this general cross-reactivity, our results are at least highlighting a major previously unreported bias in the selection of these TCRs.

Thus, polyspecificity appears to have the properties of what was described above as '*degenerate recognition of non-homologous peptides*' (*Birnbaum et al., 2014*). We prefer to describe polyspecificity as defining a fuzziness (rather than 'degeneration') in TCR recognition that makes it capable of interacting with multiple unrelated peptides and that is not strictly HLA-restricted. In this regard, it is noteworthy that B cells have a machinery for somatic mutations of their BCRs that ultimately allows them to generate antibodies with increased affinity (specificity) for antigens. While TCR generation and BCR generation share many common mechanisms, the fact that T cells did not evolve to use such somatic mutations suggest that T-cell recognition may have been selected to be more fuzzy than stringent.

As TCRs have been essentially selected during thymocyte differentiation for their ability to bind HLA molecules presented by thymic antigen-presenting cells, it could have been that polyspecificity represents some loose remnant of that property. However, it should be emphasized that a single cell does not bind randomly to any homologous-HLA/peptide complexes (*Figure 4—figure supplement 1*). Indeed, each polyspecific TCR has its own binding pattern regarding a large set of dextramers, with most that do not bind. Thus, this binding is not just '*nonspecific*' and must obey some structural rules that remain to be defined.

Polyspecific TCR binding properties are distinct from those of innate-like MAIT and NKT cells (*Toubal et al., 2019*; *Mori et al., 2016*). These have a restricted diversity, with an invariant TCRα chain and a constrained TCRβ repertoire, and are MR1- or CD1-restricted, respectively. In contrast, the repertoire of polyspecific TCRs is highly diverse for both the TCR αand β chains, although with a specific usage of VDJ/VJ recombination. Also, polyspecific T cells found in adults cannot be the remnant of fetal/neonatal T cells that have been described as having a promiscuous repertoire, as these cells have repertoires enriched in germline-encoded TCRs (*Rudd, 2020*; *Davenport et al., 2020*).

The polyspecific nature of these TCRs has a functional relevance, as we show here that cloned polyspecific TCRs indeed trigger cell activation in response to multiple peptides, although with different efficiencies. In agreement with our results, the analysis of the TCR repertoire of lung tumor-infiltrating T cells using GLIPH2, as we did here, recently identified a key TCR that, after cloning and expression, could bind and respond to a tumor-specific self-antigen as well as to a peptide from *Escherichia coli* and one from EBV (*Chiou et al., 2021*). In the same line, a recent study of multiple sclerosis pathogenesis identified autoreactive CD4+ T cell clones that can respond to self-peptides as well as peptides from EBV and from the bacteria *Akkermansis muciniphila* (*Wang et al., 2020*). This further establishes the existence of psT cells that respond to multiple antigens from different microbes. Also, in the development of the T-scan platform aimed at characterizing the specificity of TCRs (*Kula et al., 2019*), the authors performed a setup experiment with CD8 cells activated by the immunodominant NLV peptide from the pp65 protein of CMV. While T-scan could identify two hits related to NLV overlapping peptides, they also detected two hits related to the unrelated CMV I1E CMV protein. Actually, tetramer staining revealed that while 25% of the T cells in the NLV-expanded population bind NLV tetramers, 2% of the T cells bind IE1 tetramers (*Kula et al., 2019*). Moreover, the fold enrichment of the IE1-specific cells was close to that of NLV-specific cells (60 vs. 80, respectively). Thus, activation with a single peptide results in a major expansion of cells with TCRs that can also respond to an unrelated peptide. These observations match well with ours, which show that the stimulation of peripheral T cells by pp65 peptides generates cells that respond to either or both pp65 and I1E peptides (*Figure 5—figure supplement 2*).

The fact that CMV, EBV, and influenza specificities are highly represented in polyspecific binding patterns likely reflects the biased representation of these specificities in the databases (15.9%, 16.2%, and 50.6%, respectively), but also highlights that the detection of the diverse viral specificities of psT cells is largely underestimated due to the very low number of CDR3s with known viral specificities for viruses other than these. Our results suggest that polyspecific TCRs could represent at least 1/5 of the CD8 selected TCR repertoire. Altogether, our results identify and define peculiar binding properties of polyspecific TCRs. This warrants to study their physicochemical characteristics, with a special interest in solving the structure of such TCRs bound to two unrelated peptides.

## Ideas and speculations

We believe that our findings may have important implications for the study of the adaptive immune response in health, diseases, and immunotherapies. In the field of infectious diseases and vaccination, they prompt reconsideration of the paradigm of highly diverse adaptive immune repertoires providing a highly antigen-specific antiviral immune response. Indeed, there is an intrinsic weakness in the view that a highly diverse TCR repertoire is ideal to protect us from a countless repertoire of constantly evolving viruses. As best illustrated by the deadly outcome of flu in the elderly or young infants, there is a very limited amount of time for the immune system to react against rapidly replicating deadly viruses. For life-threatening situations, the initial recruitment of frequent polyspecific effector T cells might be more efficient and rapid than having to rely on rare cells with stringent specificity that would need a period of expansion to provide enough fighters. In the race between virus replication

and the mounting of the immune response, this early response of polyspecific T cells could provide some control of the viral spread that will allow time for the development of an immunologically fittest response to ensure the final control of the infection. This would explain how a very restricted repertoire of only about 1000 different TCRs arising from a single T-cell progenitor was sufficient to cope with viral infections in a child (*Bousso et al., 2000*). They would also explain why children vaccinated against measles in underdeveloped countries have a better life expectancy than unvaccinated ones (when excluding measles-related events), which could be linked to an overall better response to infections with other pathogens (*Aaby et al., 2003*; *Aaby et al., 2010*).

While our findings do not challenge that there are highly specific TCRs nor the importance of specific immune responses, they highlight another mechanism of preparedness of the immune system, reminiscent of the role of (i) other unconventional T cells like MAIT and NKT cells (*Godfrey et al., 2015*), (ii) TCR activation by bacterial superantigen (*Hayday and Vantourout, 2020*), and (iii) natural antibodies specific for microbial determinants (*Panda and Ding, 2015*).

Altogether, polyspecific TCRs form a paratopic network, made of TCRs that have a high probability of generation and are positively selected during thymocyte differentiation. These properties indicate that these TCRs have been positively selected during evolution, further supporting their overall beneficial effects. A fuzzy recognition by polyspecific TCRs would explain the so-called 'heterologous immunity' (*Welsh and Selin, 2002*; *Sewell, 2012*) in which T-cell responses to one pathogen can have a major impact on the course and outcome of a subsequent infection with an unrelated pathogen (*Furman et al., 2015*). We speculate that individual histories of fuzzy immune responses may thus create patient-specific 'antigenic sins' that might be responsible for the diverse quality of immune responses to viruses, from unapparent infection to fulminant immunopathology (*Peteranderl et al., 2016*; *Bertoletti and Ferrari, 2016*; *Tay et al., 2020*). Further studies will have to evaluate the contribution of fuzzy immune responses to the efficacy but also the immunopathology of antimicrobial responses and autoimmunity. We speculate that the origin of many immune diseases should be sought in the repeated activation of polyspecific T cells.

## Materials and methods

### Patients and samples

Thirteen human thymus samples were obtained from organ donors undergoing surgery (Department of Cardiac Surgery, Pitié-Salpêtrière Hospital, France) after approval by the *Agence de Biomédecine* and the *Ministry of Research*. Their age at the time of sampling ranged from 19 to 65 years old. The male-to-female sex ratio was 2.6.

For cross-activation experiments, six leukapheresis samples were freshly collected from healthy donors at EFS Paris Saint-Antoine-Crozatier (Etablissement Français du Sang, Paris, France) after informed consent and according to institutional guidelines. Donor selection was based on matching HLA-A2 Class I allele.

### Isolation of thymocytes and extraction of RNA

Single-cell suspensions were prepared from the thymus by mechanical disruption through nylon mesh (cell strainer). Single-cell suspensions from whole thymus were stained with antibodies anti-CD3 (AF700), anti-CD4 (APC), and anti-CD8 (FITC). Cells were sorted by fluorescent activated cell sorting (Becton Dickinson FACSAria II) with purity >95% to collect populations based on the following labeling: DPCD3$^-$ were gated as CD3$^-$CD4$^+$CD8$^+$, DPCD3$^+$ were gated as CD3$^+$CD4$^+$CD8$^+$ and CD8$^+$ were gated as CD3$^+$CD4$^-$CD8$^+$. RNA was isolated from sorted populations by means of lysis buffer with the RNAqueous-Kit (Invitrogen) extraction kit, according to the manufacturer's protocol.

### TCR repertoire library preparation and sequencing

TCR repertoire library preparation and sequencing were performed has previously described (*Barennes et al., 2021*). Briefly, TCR alpha and beta libraries were prepared on 100 ng of RNA from each sample with the SMARTer Human TCR a/b Profiling Kit (Takarabio) following the provider's protocol. Briefly, the reverse transcription was performed using TRBC reverse primers and further extended with a template-switching oligonucleotide (SMART-Seq v4). cDNAs were then amplified following two semi-nested PCRs: a first PCR with TRBC and TRAC reverse primers as well as a forward

primer hybridizing to the SMART-Seqv4 sequence added by template-switching and a second PCR targeting the PCR1 amplicons with reverse and forward primers including Illumina Indexes allowing for sample barcoding. PCR2 was then purified using AMPure beads (Beckman Coulter). The cDNA samples were quantified and their integrity was checked using DNA electrophoresis performed on an Agilent 2100 Bioanalyzer System in combination with the Agilent DNA 1000 kit, according to the manufacturer's protocol. Sequencing was performed with Hiseq 2500 (Illumina) SR-300 protocols using the LIGAN-PM Genomics platform (Lille, France).

## TCR deep sequencing data processing

FASTQ raw data files were processed for TRA and TRB sequences annotation using MiXCR (*Bolotin et al., 2015*) software (v2.1.10) with RNA-Seq parameters. MiXCR extracts TRAs and TRBs and provide corrections of PCR and sequencing errors.

## Network generation and representation

To construct a network, we computed a distance matrix of pairwise Levenshtein distances between CDR3s using the 'stringdist' (*Loo and Van der, 2014*) R package. When two sequences were similar under the defined threshold, LD>1 (i.e., at most one AA difference), they were connected and designated as 'clustered' nodes. CDR3s with more than one AA difference from any other sequences are not connected and were designated as 'dispersed' nodes.

Layout of networks for *Figures 1C and 2D* was obtained by using the graphopt algorithm of the 'Igraph' (*Csardi and Nepusz, 2006*) R package and plotted in 2D with 'ggplot2' to generate figures (*Wickham, 2009*). Only clustered nodes are represented, edges are not shown, and colors represent the node degree (log scale). Layouts of detailed networks in *Figure 1A* were done with Cytoscape (*Shannon et al., 2003*).

## Statistical analysis and visualization

Normalization was performed by sampling on the top α or β 18,000 CDR3s based on their frequency in each sample. The repertoires with less than 18,000 α or β CDR3s were not included in the statistical analysis. The numbers of samples included in the statistical analysis for the β repertoire were: 2 for DPCD3⁻, 10 for DPCD3⁺, and 12 for CD8⁺. The numbers of samples included in the statistical analysis of the α repertoire were: 6 for DPCD3⁺ and 10 for CD8⁺. Statistical tests used to analyze data are included in the figure legends. Comparisons of two groups were done using the Mann–Whitney test (*Figure 1B–D–E*, *Figure 2A–C*, *Figure 3A–B–F*) and multiple t test (*Figure 2E*). The correlation coefficient was calculated using the Pearson correlation coefficient (*Figure 2B and C*, *Figure 2C*). Enrichment of public CDR3s or virus-associated CDR3s was done using the two-tailed Chi-square test with Yates correction (*Figure 2—source data 1*, *Figure 3—source data 1*, *Figure 4—source data 1*). Statistical comparisons and multivariate analyses were performed using Prism (GraphPad Software, La Jolla, CA) and using R software version 3.5.0 (https://www.r-project.org/). PCA was performed on the frequency of VJ combination usage frequency within each donor using the factoextra R package. t-SNE was generated using the binding scores of each cell across all the antigens present in the data set. The function Rtsne of the homonymous R package (*Krijthe, 2015*) was applied with the perplexity parameter set to 10.

## Probability of generation calculation

The generation probability (*Pgen*) of a sequence is inferred using the Olga (*Sethna et al., 2019*) algorithm, which is inferred by IGoR (*Marcou et al., 2018*), for *Figure 1E* and *Figure 2B*. IGoR uses out-of-frame sequence information to infer patient-dependent models of VDJ recombination, effectively bypassing selection. From these models, the probability of a given recombination scenario can be computed. The generation probability of a sequence is then obtained by summing over all the scenarios that are compatible with it.

## CDR3 connections between individuals

In *Figure 2D*, the top 1500 βCDR3s were sampled from each of the 12 data sets of DPCD3⁺ and CD8⁺, then merged to obtain two data sets of 18,000 βCDR3s for DPCD3⁺ and CD8⁺. We generated

and represented networks, as described above, to investigate the βCDR3 inter-individual network structure.

## Virus-specific CDR3 tetramer public databases

The virus-associated CDR3 databases used for the search for specificity were compiled from the most complete previously published McPAS-TCR (*Tickotsky et al., 2017*) and VDJdb (*Shugay et al., 2018*) databases. Virus-associated βCDR3s were selected from the original data sets only when derived from a TCR of sorted CD8 T cells that were bound by a specific tetramer. A total of 5437 such unique tetramer-associated βCDR3s were identified and used. Peptides used for tetramer sorting were from cytomegalovirus (CMV), Epstein-Barr virus (EBV), hepatitis C virus, herpes simplex virus 2, human immunodeficiency virus (HIV), influenza, and yellow fever virus (YFV).

## Virus-specific CDR3 single-cell dextramer public data set

This data set contains single-cell alpha/beta TCRs from 160,914 CD8$^+$ T cells isolated from peripheral blood mononuclear cells (PBMCs) from four healthy donors. Briefly, 30 dCODE Dextramer reagents (Immudex) with antigenic peptides derived from infectious diseases (9 from CMV, 12 from EBV, 1 for influenza, 1 for HTLV, 2 for HPV, and 5 for HIV) were simultaneously used to mark cells. Each Dextramer reagent included a distinct nucleic acid barcode. A panel of fluorescently labeled antibodies was used to sort pure Dextramer-positive cells within the CD8$^+$ T-cell population using an MA900 Multi-Application Cell Sorter (Sony Biotechnology) in a reaction mix containing RT Reagent Mix and Poly dT RT primers. The Chromium Single Cell V(D)J workflow generates single cell V(D)J and Dextramer libraries from amplified DNA derived from Dextramer-conjugated barcode oligonucleotides, which are bound to TCRs. Chromium Single Cell V(D)J enriched libraries and cell surface protein libraries were quantified, normalized, and sequenced according to the user guide for Chromium Single Cell V(D)J reagent kits with feature barcoding technology for cell surface protein. We used this data set to study the presence of multiple specificities in TCR and CDR3. There were 139,378 unambiguous TCRs (with only one α and one β chain). We set the threshold defining positive binding at UMI counts greater than 10 for any given dextramer. This identified 15,195 unique virus-specific TCRs with at least one binding.

## Cross-activation experiment

PBMCs were separated on Ficoll gradient. CD8$^+$ T cells were isolated from PBMCs by positive isolation using the DYNABEADS CD8 Positive Isolation Kit (Thermo Fisher Scientific) according to the manufacturer's instructions. emCD8 T cells were purified after staining with CD3-AF700, CD8-KO, and CD45RA-PeCy7 according to the manufacturer's instructions. The samples were also stained either with CMV pp65 NLVPMVATV or with EBV BMLF-1 GLCTLVAML PE-conjugated Dextramers (Immudex). emCD8$^+$Dex$^+$ cells were sorted by FACS (FACS Aria II; BD Biosciences) with a purity >95%. Sorted cells were cultured at a maximum of 5×10$^5$ cells/mL in round-bottom 96-well plates in RPMI 1640 medium supplemented with 10% FCS, 1% penicillin/streptomycin, and glutamate at 37°C with 5% CO$_2$. In vitro stimulation was performed 24 hr after cell sorting. Sorted cells were stimulated for 6 hr with either nothing or 1 μg/mL of SIINFEKL ovalbumin peptide (OVA), NLVPMVATV CMV pp65 peptide (CMV), or GLCTLVAML EBV BMLF-1 peptide (Ozyme). The positive control (Ctl PMA/Iono) was performed with 50 ng/mL phorbol myristate acetate (PMA) and 1 mM ionomycin. Intracellular IFN-γ production with an IFN-γ-FITC antibody (BD Pharmingen) was detected in the presence of Golgi-Plug (BD Pharmingen) after fixation and permeabilization (BD Cytofix/Cytoperm). Data were acquired using a Navios flow cytometer and analyzed with Kaluza analysis software (Beckman Coulter).

## In vitro CMVpp65 CTL selection

PBMCs were obtained from blood donors at the Etablissement Français du Sang (EFS) with informed consent (Blood products transfer agreement relating to biomedical research protocol 97/5-B—DAF 03/4868). CMV$^+$ donor PBMCs were resuspended in the serum-free culture medium TexMacs (Miltenyi-Biotech) at 10$^7$ cells/mL and at a density of 5×10$^6$ cells/cm$^2$. Then, 20 μL PepTivator-CMVpp65 (Miltenyi-Biotech; 130-093-435), 30 nmol of each peptide, and approximately 140 peptides were added per mL of cell suspension. The final concentration of PepTivator-CMVpp65 was 0.6 nmol (approximately 1 μg of each peptide/mL). With these stimulation conditions, PBMCs

were incubated for 4 hr and then transferred to culture flasks at $10^6$ cells/mL in RPMI medium with 8% human serum (EFS Pays de la Loire, France) and 50 IU IL-2/mL. CTL was maintained in culture for 25 days.

## Detection of CMV pp65-specific T cells by intracellular staining with anti-INF-γ-PE

CMVpp65 selected CTL were stimulated for 6 hr in TexMacs medium (without human serum and IL-2) with 20 μL PepTivator-CMVpp65 per mL of cell suspension. Brefeldin-A was added at 10 μg/mL. Negative control (CTL without stimulation) and positive control (CTL stimulated with PMA-ionomycin) were also included. Stimulation with irrelevant peptides was also performed with PepTivator-CMV IE-1 (Miltenyi; 130-093-494) and PepTivator-EBV BZLF1 (Miltenyi; 130-093-612). Cells were then stained with anti-CD8-FITC, fixed, permeabilized, and intracellularly stained with anti-INF-γ-PE.

## Peptide dose-response of TCR transductants

TCR transductants were generated as described (*Mann et al., 2020*). Briefly, 5KC T-hybridoma cells (kindly provided by M. Nakayama, Barbara Davis Center, Aurora, CO) were transduced with the NFAT-driven fluorescent reporter ZsGreen-1 along with human CD8 by spinoculation with retroviral super-natant produced from phoenix-eco cells (ATCC CRL-3214). These cells were subsequently transduced with retroviral vectors encoding a chimeric TCR alpha gene followed by a porcine teschovirus-1 2A (P2A) peptide and a chimeric TCR beta gene synthesized by TwistBioscience. The list of transduced αβTCRs is provided in *Figure 5—source data 1*. K562 cells (ATCC CCL-243) were transduced with lentiviral vectors (Takara Bio) encoding HLA- A*02:01 or HLA-A*03:01 by spinoculation with ultracen-trifuge concentrated viral particles produced from 293T-HEK-TN cells (System Biosciences), followed by sorting of cells stained with anti-human HLA-A, B, and C antibody conjugated with PE-Cy7 (clone W6/32, BioLegend).

TCR transductants (20,000 cells per well) were stimulated for 18 hr with individual peptides at different concentrations (9- to 10-fold serial dilutions from 100 μM) in the presence of HLA- A*02:01+or HLA-A*03:01+K562 cells (50,000 cells per well), followed by analysis of ZsGreen-1 expression. Peptides (from Synpeptide) tested for the response by TCRs are GILGFVFTL (Flu MP 58–66), GLCTL-VAML (EBV BMLF1 280–288), KLGGALQAK (CMV IE1 280–288), VMNILLQYV (GAD65 114–122), VAANIVLTV (ZnT8 186–194), and other sequences selected by testing a GAD 114–122-reactive TCR by combinatorial peptide libraries, as described (*Mann et al., 2020*) EC50 values were calculated using the nonlinear regression log (agonist) versus response (three parameters) equation model in GraphPad Prism 9.

## Data and materials availability

Data sets from VDJdb were downloaded from https://vdjdb.cdr3.net. Data sets from McPAS-TCR were downloaded from http://friedmanlab.weizmann.ac.il/McPAS-TCR/. We manually curated these data sets to be sure to use only βCDR3s from CD8 tetramer-specific cells.

Single-cell data sets from 10× genomics were downloaded from https://support.10xgenomics.com/single-cell-vdj/datasets ('Application Note—A New Way of Exploring Immunity' section, data sets 'CD8+ T cells of Healthy Donor' 1–4, available under the Creative Commons Attribution license).

Data from the organ donors for thymic DP and CD8SP repertoires have been uploaded on NCBI with the BioProject ID PRJNA683011.

## Acknowledgements

The authors would like to express their gratitude to the organ donors and their families who allowed the collection of samples for research under sad circumstances. The authors would also like to thank Prof. Pascal Leprince, Dr. Guillaume Lebreton, and Dr. Marina Rigolet of the cardiac surgery team, Prof. Bruno Riou and the graft coordination team of the Pitié-Salpêtrière hospital for their invaluable contribution to sample collection. The authors thank the UMR 8199 LIGAN-PM Genomics platform (Lille, France) for efficient sequencing. The authors also thank Thierry Mora, and Aleksandra Walczak for helpful discussion.

## Additional information

### Competing interests

Valentin Quiniou: Valentin Quiniou is affiliated with Parean biotechnologies. The author has no financial interests to declare. Hang-Phuong Pham: Hang-Phuong Pham is affiliated with ILTOO pharma and Parean biotechnologies. The author hasno financial interests to declare. The other authors declare that no competing interests exist.

### Funding

| Funder | Grant reference number | Author |
|---|---|---|
| European Research Council | FP7-IDEAS-ERC-322856, TRiPoD | David Klatzmann |
| Agence Nationale de la Recherche | ANR-11- IDEX-0004-02 | David Klatzmann |
| Fondation pour la Recherche Médicale | | Roberto Mallone |
| Juvenile Diabetes Research Foundation United States of America | | Zhicheng Zhou |
| Agence Nationale de la Recherche | LabEx Transimmunom | David Klatzmann |
| Agence Nationale de la Recherche | ANR-16- RHUS- 0001 | David Klatzmann |
| Agence Nationale de la Recherche | RHU IMAP | David Klatzmann |

The funders had no role in study design, data collection and interpretation, or the decision to submit the work for publication.

### Author contributions

Valentin Quiniou, Resources, Data curation, Formal analysis, Validation, Investigation, Visualization, Methodology, Writing – original draft; Pierre Barennes, Data curation, Formal analysis, Visualization, Methodology, Writing – review and editing; Vanessa Mhanna, Formal analysis, Visualization, Methodology, Writing – review and editing; Paul Stys, Helene Vantomme, Federica Martina, Mikhail Shugay, Adrien Six, Formal analysis; Zhicheng Zhou, Formal analysis, Methodology; Nicolas Coatnoan, Investigation; Michele Barbie, Béatrice Clémenceau, Barbara Brandao, Methodology; Hang-Phuong Pham, Formal analysis, Visualization; Henri Vie, Supervision, Investigation; Roberto Mallone, Supervision, Funding acquisition, Writing – review and editing; Encarnita Mariotti-Ferrandiz, Formal analysis, Validation, Visualization, Writing – review and editing; David Klatzmann, Conceptualization, Resources, Formal analysis, Supervision, Funding acquisition, Validation, Investigation, Visualization, Methodology, Writing – original draft, Project administration, Writing – review and editing

### Author ORCIDs

Valentin Quiniou  http://orcid.org/0000-0002-0792-341X
Vanessa Mhanna  http://orcid.org/0000-0002-9354-7210
Zhicheng Zhou  http://orcid.org/0000-0001-9137-7951
Mikhail Shugay  http://orcid.org/0000-0001-7826-7942
David Klatzmann  http://orcid.org/0000-0002-0054-3422

### Ethics

Human thymus samples were obtained from organ donors undergoing surgery (Department of Cardiac Surgery,Pitié-Salpêtrière Hospital, France) after approval by the Agence de Biomédecine and the Ministry of Research, authorization n°2014-108.Leukapheresis samples were freshly collected from healthy donors at EFS Paris Saint-Antoine-Crozatier (Etablissement Français du Sang, Paris, France) after informed consent and according to institutional guidelines.

Decision letter and Author response
Decision letter https://doi.org/10.7554/eLife.81274.sa1
Author response https://doi.org/10.7554/eLife.81274.sa2

## Additional files

### Supplementary files
• MDAR checklist

### Data availability

Data sets from VDJdb were downloaded from https://vdjdb.cdr3.net. Data sets from McPAS-TCR were downloaded from http://friedmanlab.weizmann.ac.il/McPAS-TCR/. We manually curated these data sets to be sure to use only βCDR3s from CD8 tetramer-specific cells. Single-cell data sets from 10X× genomics were downloaded from https://support.10xgenomics.com/single-cell-vdj/datasets ('Application Note - —A New Way of Exploring Immunity' section, data sets 'CD8+ T cells of Healthy Donor' 1–4, available under the Creative Commons Attribution license). Data from the organ donors for thymic DP and CD8SP repertoires have been uploaded on NCBI with the BioProject ID PRJNA683011.

The following dataset was generated:

| Author(s) | Year | Dataset title | Dataset URL | Database and Identifier |
|---|---|---|---|---|
| Valentin D, David K | 2023 | human thymocytes | http://www.ncbi.nlm.nih.gov/bioproject/PRJNA683011 | NCBI BioProject, PRJNA683011 |

The following previously published datasets were used:

| Author(s) | Year | Dataset title | Dataset URL | Database and Identifier |
|---|---|---|---|---|
| 10X Genomics | 2019 | CD8+ T cells of Healthy Donor 1 | https://www.10xgenomics.com/resources/datasets/cd-8-plus-t-cells-of-healthy-donor-1-1-standard-3-0-2 | 10X Genomics, cd-8-plus-t-cells-of-healthy-donor-1-1-standard-3-0-2 |
| 10X Genomics | 2019 | CD8+ T cells of Healthy Donor 2 | https://www.10xgenomics.com/resources/datasets/cd-8-plus-t-cells-of-healthy-donor-2-1-standard-3-0-2 | 10X Genomics, cd-8-plus-t-cells-of-healthy-donor-2-1-standard-3-0-2 |
| 10X Genomics | 2019 | CD8+ T cells of Healthy Donor 3 | https://www.10xgenomics.com/resources/datasets/cd-8-plus-t-cells-of-healthy-donor-3-1-standard-3-0-2 | 10X Genomics, cd-8-plus-t-cells-of-healthy-donor-3-1-standard-3-0-2 |
| 10X Genomics | 2019 | CD8+ T cells of Healthy Donor 4 | https://www.10xgenomics.com/resources/datasets/cd-8-plus-t-cells-of-healthy-donor-4-1-standard-3-0-2 | 10X Genomics, cd-8-plus-t-cells-of-healthy-donor-4-1-standard-3-0-2 |

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
