## [Editor Report]

This manuscript reports novel and valuable observations regarding human CD8 T cells that express shared T cell receptors amongst individuals and exhibit poly-specificity directed mainly to several unrelated viral antigens. In the revised version, the authors have addressed the majority of objection raised by the reviewers and the additional results and revised discussion provide a solid support to the authors claims. The results from these studies will add to the ongoing debate on T cell specificity by providing material for an informed decision on whether the data represent genuine cross-reactivity or technical confounders.

---

## [Decision Letter]

**Decision letter after peer review:**

Thank you for submitting your article "Human thymopoiesis produces polyspecific CD8^+^ α/β T cells responding to multiple viral antigens" for consideration by *eLife*. Your article has been reviewed by 3 peer reviewers, and the evaluation has been overseen by a Reviewing Editor and Tadatsugu Taniguchi as the Senior Editor. The following individual involved in review of your submission has agreed to reveal their identity: Benny Chain (Reviewer #2).

Essential revisions:

1. Please comment on the correlation between TCR clustering, high Pgen, and sharing of TCRs amongst individuals.

2. Please address the issue concerning the lack of clarity in sections describing the relationship between thymic selection and poly-specificity, and between in silico "cross-reactivity" as evidenced by multiple annotations and the functional poly-specific T cells.

3. The mechanistic molecular details underlying poly-specificity also remain unclear. For one, the evidence for large-scale cross-reactivity comes mainly from a set of dextramers for A*03 and A*11-restricted peptides. But these dextramers appear to be binding in a uniquely non-specific and non-TCR-dependent manner in this experiment. Hence, TCR sequence doesn't appear to be the determining factor for binding to these dextramers; rather it may be expression of KIR genes or other surface proteins that can interact with MHC.

4. The claim that clustered public CDR3s are enriched in viral specificities is not justified by the data, which comes from sequence matching against literature-derived databases. To maintain this claim, it is necessary to show experimental specificity data on the very same datasets to make a conclusion about viral specificities in general.

5. In your discussion, please consider the observation of Slabodkin and colleagues who have shown that generation models differ by individual (Genome Research 2021). If that were true for TCR repertoires, how would this change your conclusions/data analysis?

6. The data need to be made public. The sequences of the individual TCRs that were expressed and characterized should be given, at least their V/J genes and CDR3 regions.

7. Figure 4., especially panels B and C, very hard to understand. Please revise the legends. What is the heatmap in B showing? What is the basis for the t-sne plot?

8. Figure S10 is missing.

9. The references for VDJdb and MCPas are incorrectly cited (wrong numbers) in multiple figure legends.

*Reviewer #1 (Recommendations for the authors):*

– "In this regard, it is noteworthy that B cells have a machinery for somatic mutations of their BCRs that ultimately allows them to generate antibodies with increased affinity (specificity) for antigens. While TCR generation and BCR generation share many common mechanisms, the fact that T cells did not evolve to use such somatic mutations suggest that T cell recognition has been selected to be more fuzzy than stringent."  I would be careful with this assessment. Antibody responses are also known to be very polyreactive. Furthermore, in order to enter the GC reaction, naive B cells must already possess affinity to the antigen. The GC does not create antigen specificity de novo. So, it may be likely that B cells underlie similar principles. It would be interesting to understand further the differences in reactivity between B and T cell responses.

– Recently, Slabodkin and colleagues have shown that generation models differ by individual (Genome Research 2021). If that were true for TCR repertoires, how would this change your conclusions/data analysis? Can you discuss this in the discussion?

*Reviewer #2 (Recommendations for the authors):*

In part 1, I think the authors need to acknowledge that publicity, pGEN, and also sequence clustering are not independent processes, but all highly linked. High pGEN T CRs exist in a part of TCR space with many neighbours. This has been discussed previously in some detail, especially by Nit Freidman and his colleagues.

I found Figure 4., especially panels B and C, very hard to understand, and the Legends were unhelpful here. What is the heatmap in B actually showing? And what is the basis for the t-sne plot? I don't understand why this plot is being used, or what it claims to show.

As mentioned above, I found the different sections of the paper not really linked. For example, it was not clear to me that functionally poly-specific T cells are enriched during positive/negative selection. But I appreciate that this may require a lot of further work, and I would be happy for the authors to be a bit more guarded in their conclusions, and try and indicate clearly the gaps in their analysis which need addressing in future studies. The Discussion is rather uncritical in this respect, which is a pity because there are lots of interesting things to say.

*Reviewer #3 (Recommendations for the authors):*

Figure S10 is missing.

The references for VDJdb and MCPas are incorrectly cited (wrong numbers) in multiple figure legends.

No data is made public, not the sequencing results nor the derivative analyses of the public databases. This should be done or it should be clarified (with justification) that privacy concerns do not allow deposition of the donor TCR sequences in public databases (which seems a little surprising). The sequences of the individual TCRs that were expressed and characterized should be given, at least their V/J genes and CDR3 regions.

[Editors' note: further revisions were suggested prior to acceptance, as described below.]

Thank you for resubmitting your work entitled "Human thymopoiesis produces polyspecific CD8^+^ α/β T cells responding to multiple viral antigens" for further consideration by *eLife*. Your revised article has been evaluated by Tadatsugu Taniguchi (Senior Editor) and a Reviewing Editor.

The manuscript has been improved but there are some remaining issues that need to be addressed, as outlined below:

1) Some revisions or corrections of the 10X dataset need to be provided as all reviewers agreed that a nonspecific dextramer binding is being detected.

2) A more thorough discussion regarding this point of needs to be included.

*Reviewer #1 (Recommendations for the authors):*

The authors have addressed all of my concerns.

*Reviewer #2 (Recommendations for the authors):*

The authors have addressed most of the key reviewer points. In particular, they document and discuss the relationships between pGen and clustering; they improve the flow between the parts of the manuscript; they have removed or clarified figures which were hard to understand; and they have somewhat softened their conclusions. As I wrote previously, I think the findings of this study are not the whole story, but contribute some interesting and provocative findings.

*Reviewer #3 (Recommendations for the authors):*

I appreciate the authors' willingness to keep an open mind in revisiting their conclusions. I remain convinced that the binding of the A*03 and A*11 dextramers in the large public 10x dextramer dataset is largely not TCR specific (except a few A*11-positive clonotypes in the A*11 positive donor 1, more below). I've attached two figures that show the distributions of dextramer UMI (barcode) counts for the 30 largest clonotypes in donors 1 and 2. As I discuss in detail below, these show pretty convincingly that the binding of A*03 and A*11 dextramers is qualitatively different from the "specific" dextramers and does not look to be mediated by TCR sequence. Now that the authors have provided the TCR sequences for the in vitro TCR reconstruction assays, I also have more questions about that data. Here are some points for the authors to consider:

– On the question of A*03/A*11 non-specific binding, the authors state "If it would be an MHC to KIR binding, then such dextramers should bind to most cells, independently of their TCRs". This ignores the fact that the cells in this dataset have diverse phenotypes (from naive to memory) and diverse transcriptional and presumably surface protein expression levels. So the fact that not all cells are bound does not imply that it is TCR mediated: these cells differ in more than just their TCRs.

– Looking at the distributions of dextramer UMI counts for donor 2, (see Author response image 1), one can see that for the "specific" dextramers (the bottom 3 rows: A02-GIL, B08-RAK, and A02-GLC) there is a very clear separation between true binding clonotypes, which show high dextramer binding for essentially all of the member cells, and non-binding clonotypes, for which the majority of cells have a dextramer UMI count of 0. Examples of true-positive clonotypes are #12,20,26,27,30 for A02-GIL; #1,2,5,10,14,16,19,21-24,28,29 for B08-RAK, and #7 for A02-GLC. For the non-specific dextramers, it's a completely different story. Each clonotype has a broad distribution of UMIs, with many cells having 0 bound dextramers, and many having multiple bound dextramers, including many that exceed the authors' threshold for dextramer positivity (the red dashed line, the count at or above the threshold is shown in red above each clone's violin). Consider clone #2, for which 4270 of the 4274 cells meet the binding threshold for B08-RAK. This clone also has 574 cells that meet the threshold for the non-specific dextramer A03-KLG, but also a large number with 0 A03-KLG dextramers bound. Would the authors say that this TCR binds to the A03-KLG peptide-MHC? How about the other A03 and A11 dextramers, which also show large numbers of cells exceeding the threshold?

**Decision letter image 1. sa1fig1:** 

– The distributions of dextramer UMI counts for donor 1, (see Author response image 2), show a very similar story, with the added feature that now we can see, for the A*11 dextramers, a mix of specific and non-specific binding. This donor is HLA-A*11 positive, and there are some genuine A*11-specific clonotypes, like the CASSLYSATGELFF clonotype that the authors put forward as proof that these dextramers bind specifically. For A*11-IVT, the true specific clonotypes are #1,11,26; for A*11-AVF they are #2 and 18 (which may in fact be the same dual-α clonotype that was inadvertently split; these both have the CASSLYSATGFELFF CDR3beta). In a sense, these A*11 dextramers in donor 1 are the "exceptions that prove the rule", because they show side-by-side the difference between non-specific UMI distributions and specific UMI distributions for the same dextramer in the same donor. Note that all the A03-KLG distributions in both donors look like non-specific distributions. I would suggest that the authors, in the interest of transparency, show a supplemental figure like these to allow readers to judge for themselves whether these dextramers are binding in a specific, TCR-sequence-dependent manner.

**Decision letter image 2. sa1fig2:** 

– The two TCRs selected for in vitro reconstitution are shown with red boxes in Author response image 1. We can see that these are both "specifically" bound by A02-GIL (Flu M158). They both also show multiple cells that are positive for the A03-KLG dextramer, including many that meet the authors' positivity threshold (28 cells for TCR#35-13 and 42 cells for TCR#36-150). The first thing to point out about the in vitro data is that neither TCR shows any response to the A03-KLG pMHC, despite the many A03-KLG dextramer-positive cells in the single-cell dataset. This is not mentioned in the manuscript but should be along with some explanation for the discrepancy, as it calls into question the validity of the A03/A11 dextramer binding. A second interesting thing about TCR#35-13 is that the other peptide it showed a response to, albeit 3-4 logs lower than the cognate flu peptide, is A02-KLGGALQAK. This is interesting because that peptide is not predicted to bind to A02 (it's well above even the "weak binder" threshold for netmhcpan, for example, which is not surprising given the lysine at the second anchor position). For TCR#36-150, what is surprising is that the strongest binding (again by 3-4 logs) is to the non-cognate A02-GLC (EBV BMLF1) pMHC, rather than the A02-GIL flu peptide. I would strongly encourage the authors to revisit this data and double-check the experiment, because (1) this TCR is just about the most canonical A02-GIL (Flu M158) TCR sequence you can possibly imagine (TRBV19, RS motif, TRAV27/TRAJ42), so it's hard to believe that it would bind so much more weakly than TCR#35-13. And (2) the dextramer data from the 10x experiment shows zero binding to A02-GLC, the supposed tight binder (Author response image 1, clone #30, far right red box), and instead strong binding to A02-GIL. And the A02-GLC dextramer did indeed work in the 10x experiment: clone #7 shows strong binding (757/772 cells meet the threshold), and many other clonotypes outside the top 30 also showed specific binding (plus the TCR repertoires for A02-GLC are consistent with previously published repertoire data for this pMHC). This inconsistency between the in vitro data and the 10x dextramer data, like the failure to see A03-KLG binding, should also be pointed out.– In their response, the authors state "we compared the CDR3 length distribution between DP CD3+ cells and CD8 SP cells from our thymic dataset. We did not observe major changes. The distribution and the mean of CDR3 length for the two cell populations remained identical." In fact, the orange distribution is significantly shifted (as the authors show) toward shorter lengths and probably also slightly compressed. It's certainly not the case that the "distribution and mean" remain identical. Perhaps the authors meant to say the "mode"? Still, the distribution is certainly not the same, and seemingly modest shifts in CDR3 length can have significant impacts on neighbor distributions.

– In Figure 4C, I wonder whether the authors are taking consistency within expanded clonotypes into account. Is a TCR called as positive for a given dextramer if any cells expressing it are bound by that dextramer, regardless of the fraction those cells represent of the full clonotype? In Author response image 1, we can see that even for the specific dextramers, there are occasional cells (typically less than 1 percent of the clone) in the non-binding clonotypes that meet the dextramer UMI threshold. There are all sorts of reasons why we might see this: minor stickiness in the reagents, non-specific dextramer adherence, incomplete washing of non-bound dextramers, incorrect TCR assignment (this happens a fair amount, possibly from ambient TCR transcripts that become trapped in the wrong emulsion).

To reiterate, I appreciate that the authors are keeping an open mind about their conclusions. I personally feel that the data as presented and as it regards the public 10x dextramer dataset needs to be corrected or revised along with the discussion on this point.

---

## [Author Response]

Essential revisions:1. Please comment on the correlation between TCR clustering, high Pgen, and sharing of TCRs amongst individuals.

We added 4 supplementary figures (Figure 1—figure supplement 3C-E-F and Figure 2 —figure supplement 1C-D) and discussed this point in the response to reviewers and the manuscript.

2. Please address the issue concerning the lack of clarity in sections describing the relationship between thymic selection and poly-specificity, and between in silico "cross-reactivity" as evidenced by multiple annotations and the functional poly-specific T cells.

We have reworked the titles of the different sections in Results to emphasize the switch from thymocyte bulk sequencing studies to single peripheral cell studies.

3. The mechanistic molecular details underlying poly-specificity also remain unclear. For one, the evidence for large-scale cross-reactivity comes mainly from a set of dextramers for A*03 and A*11-restricted peptides. But these dextramers appear to be binding in a uniquely non-specific and non-TCR-dependent manner in this experiment. Hence, TCR sequence doesn't appear to be the determining factor for binding to these dextramers; rather it may be expression of KIR genes or other surface proteins that can interact with MHC.

We respectfully disagree.

If it would be a KIR binding, then the dextramers would bind to most cells independently of their TCRs. We have added two supplementary figures to emphasize that this is not the case (Figure 4 —figure supplement 1B-C). If it would be a binding to “other surface proteins”, it would likely be the same.

4. The claim that clustered public CDR3s are enriched in viral specificities is not justified by the data, which comes from sequence matching against literature-derived databases. To maintain this claim, it is necessary to show experimental specificity data on the very same datasets to make a conclusion about viral specificities in general.

“Sequence matching against literature-derived databases” is a much used and accepted mean to provide indication on the specificity of clusters, for example those generated by LV distance or GLIPH2. This is the only possibility to infer specificity for bulk TCR sequences from thymocytes, as these are not from αβ paired TCR sequences. In our study, the validity of thymocyte antigen-specificity inference is further supported by the functional data obtained with αβ paired TCR sequences from single-cell sequenced peripheral blood lymphocytes that have necessarily gone through thymocyte differentiation. The reviewers have clearly stated that experimental specificity determination with thymocytes TCRs was beyond the scope of this study.

This said, we have softened our claim by stating: “The polyspecific nature of these TCRs translates in polyreactivity as we show here that cloned polyspecific TCRs from the single cell dataset indeed trigger cell activation in response to multiple peptides. Similar studies could not be performed with thymocyte TCRs as these were obtained by bulk sequencing and thus not as paired α/β TCRs. Nonetheless, the polyspecific/polyreactive TCRs from the single cell dataset necessarily originated from a successful thymocyte differentiation.”

5. In your discussion, please consider the observation of Slabodkin and colleagues who have shown that generation models differ by individual (Genome Research 2021). If that were true for TCR repertoires, how would this change your conclusions/data analysis?

We now mention this observation and quote this article (Ref 28 in the revised version).

6. The data need to be made public.

They will of course be made public when the manuscript is published. They actually have already been loaded. The NCBI project ID has been indicated in the revised version.

The sequences of the individual TCRs that were expressed and characterized should be given, at least their V/J genes and CDR3 regions.

We now provide these sequences as a supplementary Table 4.

7. Figure 4., especially panels B and C, very hard to understand. Please revise the legends. What is the heatmap in B showing? What is the basis for the t-sne plot?

We have deleted the panel B that is not essential and is indeed hard to understand, and we have improved the figure 4 legend.

8. Figure S10 is missing.

This was an oversight that has been corrected.

9. The references for VDJdb and MCPas are incorrectly cited (wrong numbers) in multiple figure legends.

This was a mistake that has been corrected.

Reviewer #1 (Recommendations for the authors):– "In this regard, it is noteworthy that B cells have a machinery for somatic mutations of their BCRs that ultimately allows them to generate antibodies with increased affinity (specificity) for antigens. While TCR generation and BCR generation share many common mechanisms, the fact that T cells did not evolve to use such somatic mutations suggest that T cell recognition has been selected to be more fuzzy than stringent."  I would be careful with this assessment. Antibody responses are also known to be very polyreactive. Furthermore, in order to enter the GC reaction, naive B cells must already possess affinity to the antigen. The GC does not create antigen specificity de novo. So, it may be likely that B cells underlie similar principles.

In the context of our observation, it is striking to us that while “TCR generation and BCR generation share many common mechanisms, T cells did not evolve to use such somatic mutations”. Indeed, there might be populations with polyreactive TCR or BCR in both T and B cells, respectively. However, it remains that T cells have no way to turn such receptors into ones with a higher affinity for a given antigen, while B cells have. We believe that this statement is just factual.

We however agree that “While TCR generation and BCR generation share many common mechanisms, T cells did not evolve to use such somatic mutations suggest that T cell recognition has been selected to be more fuzzy than stringent " is speculative and we thus replaced “has been selected” by “may have been selected”

It would be interesting to understand further the differences in reactivity between B and T cell responses.

We totally agree and intend to continue such studies.

– Recently, Slabodkin and colleagues have shown that generation models differ by individual (Genome Research 2021). If that were true for TCR repertoires, how would this change your conclusions/data analysis? Can you discuss this in the discussion?

This is a relevant observation that we now quote in the manuscript.

If this would be true for TCR repertoires, it would not change our current views. Indeed, as discussed above, we believe that, for example, the co-clustering between individuals is more important than the slight differences in Pgen.

Reviewer #2 (Recommendations for the authors):In part 1, I think the authors need to acknowledge that publicity, pGEN, and also sequence clustering are not independent processes, but all highly linked.

Indeed, there is a significant positive correlation between the Pgen and the number of connections among the clustered TCRs, as was reported in Figure 1F of the original manuscript. Furthermore, this correlation is true for both private and public TCRs, as was reported in figure 2B of the original manuscript.

To show the link between the three phenomena, we now have added two supplementary figures (Figure 2 —figure supplement 1C and D) showing a high positive correlation between Pgen and the number of connections, and between cross-individual sharing and the number of connections, and to a lesser extent between Pgen and cross-individual sharing.

However, we would like to emphasize that the difference in the mean Pgen of the clustered and dispersed TCRs is of about 20-fold. This is a high difference for a biological process (and highly statistically significant), but a small one compared to the 10^-log10^ span of the Pgen of the two populations. Factually, what we observed is not that clustered sequences have a high Pgen, but that they have a higher Pgen than the non-clustered sequences (New Figure 1—figure supplement 3E). Yet, many CDR3s with high Pgen do not cluster (Figure 1—figure supplement 3*F)* indicating that a high Pgen is not the only (nor most important) driver of clustering.

High pGEN T CRs exist in a part of TCR space with many neighbours. This has been discussed previously in some detail, especially by Nit Freidman and his colleagues.

Indeed, we are aware of and were actually quoting the "T cell receptor repertoires of mice and humans are clustered in similarity networks around conserved public CDR3 sequences" article published in 2017 by Friedman and colleagues (our reference n°22). Nonetheless, their analyses focused on peripheral CD4^+^ T cells *from mouse* (although they also briefly looked at CD8^+^ cells). An analysis of *human* DPCD3-, DPCD3+ and CD8^+^ from the thymus has, to our knowledge, never been previously reported. Our results are in line with those of Friedman and colleagues. They shed more light on these observations from a repertoire generation and thymic selection perspective that was not addressed by these authors.

I found Figure 4., especially panels B and C, very hard to understand, and the Legends were unhelpful here.

We agree that the figures are complex, as the subject is. We have decided to delete Figure 4B that does not add much and is indeed difficult to understand and we improved the rest of the legend the figure 4 legend to clarify it.

What is the heatmap in B actually showing?

For 4.B., an improved legend should have read: “In silico analysis of EBV specific CDR3β. Using GLIPH2, we clustered a data set made of the aggregation of CDR3s from the 10x single cell dataset and our ThyCD8 dataset. The clusters “specificities” were then inferred based on them containing an EBV-specific CDR3 as defined from the single cell data. For each cluster, the counts of each of their CDR3 within CD8^+^ thymocytes were used to compute the Morisita-Horn index between the CDR3 from these different clusters. The heatmap represent the MH for patients P18 and P25.

The high Morisita-Horn index highlight similarities in the CDR3s from clusters recognizing distinct EBV antigens, and, noteworthy, similarly for the two patients studied.

This figure now has been deleted.

And what is the basis for the t-sne plot? I don't understand why this plot is being used, or what it claims to show.

For 4.C. T-sne plot of CDR3beta sequences from peripheral CD8 cells based on sequence similarity. On this overall repertoire representation, we labeled the cells that were annotated as EBV specific (blue, Left) and CMV specific (green, Right). The side-by-side analysis reveals that some of those labelled cells are both labelled in blue and green, therefore that they are binding both EBV and CMV dextramer.

As mentioned above, I found the different sections of the paper not really linked.

To make this link clearer, we have reworked the titles of the different Results’ sections to emphasize the switch from thymocyte bulk sequencing studies to that of single peripheral cell sequencing studies.

For example, it was not clear to me that functionally poly-specific T cells are enriched during positive/negative selection. But I appreciate that this may require a lot of further work,

Indeed, the evidences for a functional polyspecificity have been made from TCRs from peripheral blood cells (that however were once thymocytes). To similarly evidence that “functionally polyspecific T cells are enriched during positive/negative selection” would require to perform single-cell analysis on the DP and CD8SP subsets, identifying multiple relevant αβ pairs, cloning them and expressing them before functionally assessing their polyreactivity, which represent a massive work.

We have softened our claim by stating: “The polyspecific nature of these TCRs translates in polyreactivity as we show here that cloned polyspecific TCRs from the single cell dataset indeed trigger cell activation in response to multiple peptides. Similar studies could not be performed with thymocytes TCRs as these were obtained by bulk sequencing and thus not as paired ab TCRs. Nonetheless, the polyspecific/polyreactive TCRs from the single cell dataset necessarily originated from a successful thymocyte differentiation.”

and I would be happy for the authors to be a bit more guarded in their conclusions, and try and indicate clearly the gaps in their analysis which need addressing in future studies. The Discussion is rather uncritical in this respect, which is a pity because there are lots of interesting things to say.

We have now modified the Discussion to be more guarded.

Reviewer #3 (Recommendations for the authors):Figure S10 is missing.

We apologize for this.

We now included the figure in the supplementary dataset.

The references for VDJdb and MCPas are incorrectly cited (wrong numbers) in multiple figure legends.

We apologize for this. Now corrected in the revised version.

No data is made public, not the sequencing results nor the derivative analyses of the public databases. This should be done or it should be clarified (with justification) that privacy concerns do not allow deposition of the donor TCR sequences in public databases (which seems a little surprising). The sequences of the individual TCRs that were expressed and characterized should be given, at least their V/J genes and CDR3 regions.

This information is now in the manuscript.

[Editors' note: further revisions were suggested prior to acceptance, as described below.]

The manuscript has been improved but there are some remaining issues that need to be addressed, as outlined below:1) Some revisions or corrections of the 10X dataset need to be provided as all reviewers agreed that a nonspecific dextramer binding is being detected.

We have carefully looked at the figures generated by reviewer-3.

We now rule out a role for KIRs and we bring evidences that the binding we detect is TCR mediated. Indeed, the “non-specific” binding patterns are both HLA and peptide dependent, and are competed for by high affinity dextramers.

2) A more thorough discussion regarding this point of needs to be included.

We have modified the discussion to highlight these new findings and interpretation.

Overall, as mentioned by the editors, these comments only concern the interpretation of binding patterns relative to the 10X dataset, one experiment from our results. Polyreactivity is robustly supported by our functional experiments.

We have added a new figure-4, a corrected Figure-5C, a new Fig-1 supp Figure 3-C and a new Fig-4 supp Figure 2A-E in our revised manuscript and made some changes in our presentation and discussion of the results.

We believe that, at this stage, it is to the readers to make their own interpretation of our results, in line with the *eLife* philosophy of reviews that led to changing its reviewing process.

Reviewer #3 (Recommendations for the authors):I appreciate the authors' willingness to keep an open mind in revisiting their conclusions.

We thank the reviewer for highlighting our open mind.

We hope to stimulate the same openness with our response to his comments.

I remain convinced that the binding of the A*03 and A*11 dextramers in the large public 10x dextramer dataset is largely not TCR specific (except a few A*11-positive clonotypes in the A*11 positive donor 1, more below). I've attached two figures that show the distributions of dextramer UMI (barcode) counts for the 30 largest clonotypes in donors 1 and 2. As I discuss in detail below, these show pretty convincingly that the binding of A*03 and A*11 dextramers is qualitatively different from the "specific" dextramers and does not look to be mediated by TCR sequence.

We thank the reviewer for providing these interesting figures that report representation of our analyses that we had not done.

They show that, yes indeed, the binding patterns of “specific” dextramer is qualitatively different from those of other dextramers (which was already disclosed in our Figure 4—figure supplement 1, in which it was apparent that the number of dextramer bound per cells varied for the different dextramers bound).

This is an observation that should not be so surprising.

Indeed, the conditions in which these dextramers’ staining were performed need to be considered.

First, T cells were labeled with a mixture of several dextramers, each dextramer competing with others for binding to the available TCRs. In these conditions, dextramers binding with higher affinity should be favored over dextramers binding the same T cells with lower affinity.Second, binding of lower affinity should result in a dimmer dextramer staining, even in the absence of such competition.Third, the activation state of these T cells (notably resulting in a modulated TCR expression) may more substantially impact the binding of such low-affinity dextramers.Fourth, having selected the 30 more expanded clonotypes may have favored the biases described above.

Based on the reviewer’s observations, we went back to the dataset and performed the same analyses on 5 clonotypes with clear specific responses and 5 randomly selected clonotypes without such binding, see Author response image 1.

**Author response image 1. sa2fig1:** 

Looking at these binding patterns, it is obvious that each dextramer has a unique “basic” binding pattern. For example, the A02GIL has a rather “thin” pattern, while A03KLG has more of a “fat” one.Looking in more details, one can also note that:

each time a clonotype exhibit a strong specific binding for one dextramer, this lowers the binding intensity for the other dextramers, also increasing the number of clonotype with 0 binding. (see for example the A03KLG binding of the first two clonotypes). This difference is actually statistically significant (see Author response image 2 comparing the number of dextramer per cell when a specific dextramer is also bond (left) or not (right))the binding patterns of A03 and A11 are “fatter” than those of A02 or B08 dextramersFor the same A03 and A11 backbones, the binding patterns are influenced by the peptide used (for example, A03KLG is fatter than A03 RLR).

Although, as discussed above, we do not rule out that other factors could influence binding, these results show that the so called “non-specific” dextramer bindings are:

HLA dependentpeptide dependentcompeted for by high affinity binding dextramers

These results point to a TCR dependent binding.

Now that the authors have provided the TCR sequences for the in vitro TCR reconstruction assays, I also have more questions about that data. Here are some points for the authors to consider:– On the question of A*03/A*11 non-specific binding, the authors state "If it would be an MHC to KIR binding, then such dextramers should bind to most cells, independently of their TCRs". This ignores the fact that the cells in this dataset have diverse phenotypes (from naive to memory) and diverse transcriptional and presumably surface protein expression levels. So the fact that not all cells are bound does not imply that it is TCR mediated: these cells differ in more than just their TCRs.

As mentioned above, we agree that T cell activation stage may influence dextramer binding patterns. We looked at the available transcriptome data of the 10X dataset, and we can rule out a KIR binding.

These results show that, as known, CD8 T cells do not have a high expression of KIRs. Moreover, a positive KIR expression (regardless of the expression level) was only detected in 6.5% of all polyreactive cells (Figure 4—figure supplement 2).

These results show that there is no difference between monospecific and polyspecific cells for KIR expression.

These results exonerating KIRs are also in line with our response to this comment in the first review. If the HLA-A11 and HLA-A3 dextramers bind to KIRs, all HLA-A11 and HLA-A3 dextramers, should be bound to the same cell irrespective of their peptide. This is not the case. For example, on figure 4—figure supplement 1C, we show that for the cell with TRA: CADTASGTYKYIF the binding is peptide dependent as only HLA-A3 dextramers with CMV and EBV peptides are bound, while HLA-A3 dextramer harboring a cancer peptide is not. Thus, the binding is peptide dependent, which should not be the case if it was a binding to KIRs

We went further in our analyses to explore if the cell status could influence the binding.

All these new results are now presented in "Figure 4—figure supplement 2. Single cell gene expression analysis" of the manuscript.

– Looking at the distributions of dextramer UMI counts for donor 2, (see Decision letter image 1), one can see that for the "specific" dextramers (the bottom 3 rows: A02-GIL, B08-RAK, and A02-GLC) there is a very clear separation between true binding clonotypes, which show high dextramer binding for essentially all of the member cells, and non-binding clonotypes, for which the majority of cells have a dextramer UMI count of 0. Examples of true-positive clonotypes are #12,20,26,27,30 for A02-GIL; #1,2,5,10,14,16,19,21-24,28,29 for B08-RAK, and #7 for A02-GLC. For the non-specific dextramers, it's a completely different story. Each clonotype has a broad distribution of UMIs, with many cells having 0 bound dextramers, and many having multiple bound dextramers, including many that exceed the authors' threshold for dextramer positivity (the red dashed line, the count at or above the threshold is shown in red above each clone's violin).

We discussed this above.

Also, to us, the violin representation may somehow bias the visual impression relative to these bindings. We thus represented them more classically, as density distribution used for cytometry data.

**Author response image 3. sa2fig3:** multi-GIL are multi-specific clonotypes with the highest specificity being towards the A02-GIL dextramer. Multi-other refers to multi-specific clonotypes without any differentially high binding to any of the dextramers. Finally, uni-GIL are uni-specific clonotypes that exclusively bind the A02-GIL dextramer.

We observed that:

Consider clone #2, for which 4270 of the 4274 cells meet the binding threshold for B08-RAK. This clone also has 574 cells that meet the threshold for the non-specific dextramer A03-KLG, but also a large number with 0 A03-KLG dextramers bound. Would the authors say that this TCR binds to the A03-KLG peptide-MHC? How about the other A03 and A11 dextramers, which also show large numbers of cells exceeding the threshold?

Again, the conditions in which these dextramers’ staining were performed need to be taken into account. First, T cells were labeled with a mixture of several dextramers. In these conditions, it is well possible that dextramers binding with higher affinity may be favored over others, e.g. cross-binding dextramers binding the same T cells with lower affinity. Second, these cross-binding being of lower affinity, it would result in a dimmer dextramer staining even in the absence of such competition. Third, the activation state of these T cells (resulting in lower TCR expression) may more substantially impact the binding of such low-affinity dextramers.

We do agree that, altogether, the analysis of this dataset is difficult and that more work will be needed for a better understanding of what they represent.

Meanwhile, we also believe that they should not just be considered as irrelevant background noise.

– The distributions of dextramer UMI counts for donor 1 , (See Decision letter image 2), show a very similar story, with the added feature that now we can see, for the A*11 dextramers, a mix of specific and non-specific binding. This donor is HLA-A*11 positive, and there are some genuine A*11-specific clonotypes, like the CASSLYSATGELFF clonotype that the authors put forward as proof that these dextramers bind specifically. For A*11-IVT, the true specific clonotypes are #1,11,26; for A*11-AVF they are #2 and 18 (which may in fact be the same dual-α clonotype that was inadvertently split; these both have the CASSLYSATGFELFF CDR3beta).

If we understood well the reviewer’s comment, that “the same dual-α clonotype was inadvertently split” is just impossible. The cells are barcoded individually. If a cell expresses two α chains, it would be detected. That a chain could be missed occasionally is possible, but this would have generated a large number of clonotypes expressing both α chains.

In a sense, these A*11 dextramers in donor 1 are the "exceptions that prove the rule", because they show side-by-side the difference between non-specific UMI distributions and specific UMI distributions for the same dextramer in the same donor.

As stated above, it should be no surprise that the binding patterns for “specific” vs “polyspecific” binding are qualitatively and quantitatively different.

Note that all the A03-KLG distributions in both donors look like non-specific distributions. I would suggest that the authors, in the interest of transparency, show a supplemental figure like these to allow readers to judge for themselves whether these dextramers are binding in a specific, TCR-sequence-dependent manner.

We now do so (Figure 4E) and discuss it.

– The two TCRs selected for in vitro reconstitution are shown with red boxes in Decision letter image 1. We can see that these are both "specifically" bound by A02-GIL (Flu M158). They both also show multiple cells that are positive for the A03-KLG dextramer, including many that meet the authors' positivity threshold (28 cells for TCR#35-13 and 42 cells for TCR#36-150). The first thing to point out about the in vitro data is that neither TCR shows any response to the A03-KLG pMHC, despite the many A03-KLG dextramer-positive cells in the single-cell dataset. This is not mentioned in the manuscript but should be along with some explanation for the discrepancy, as it calls into question the validity of the A03/A11 dextramer binding.

We now discuss it. See below

A second interesting thing about TCR#35-13 is that the other peptide it showed a response to, albeit 3-4 logs lower than the cognate flu peptide, is A02-KLGGALQAK. This is interesting because that peptide is not predicted to bind to A02 (it's well above even the "weak binder" threshold for netmhcpan, for example, which is not surprising given the lysine at the second anchor position).

Indeed, we were also surprised by this reactivity of A02-KLGGALQAK, which we initially included in our experiments as a “control” for A03-KLGGALQAK. We thus verified it multiple times with the same result (also note that the response is nicely dose-dependent and saturable).

For this peptide, the NetMHCpan4.1 returns a ranking for the HLA-A2 binding of this peptide of 12.4%. Although lower than the customary 2% used to define weak binders, we have several examples of peptides with these low scores that nonetheless show some experimental binding to HLA, which is sufficiently stable to allow recombinant HLA/peptide refolding. Examples can be found in Gonzalez-Duque et al., Cell Metab 2018 (Figure S2), Azoury et al., Diabetes 2020 (Table 1 and Figure S2), Azoury et al., Diabetes 2021 (Figure 1).

Indeed, this customary 2% binding threshold has been defined mostly based on viral and tumor immunology studies that may apply less stringently to cross-reactive T-cell responses, and to the autoimmune responses analyzed in the references provided.

Finally, there is a growing literature (previously cited in our article, e.g. Ebrahimi-Nik et al., Nat Commun 2021) highlighting peptides with non-detectable HLA binding that are nonetheless capable of eliciting significant T-cell responses mediating tumor rejection in vivo.

The possible functional relevance for autoimmunity of these weak interactions has been discussed in a recent review (Samassa and Mallone, Curr Opin Endocrinol Diabetes Obes 2022), and some of these considerations may also apply to the cross-reactive T cells that we describe.

Altogether, the response to the A02 or A03 presented KLGGALQAK peptide illustrate that the Binding assays do not fully/always predict the TCR mediated activation. In this line, it is known that the TCR avidity threshold for functional responses is notoriously lower than that required for dextramer binding (see Rius et al., J Immunol 2018). We believe that the activation assay is more quantitative and robust and importantly more relevant.

For TCR#36-150, what is surprising is that the strongest binding (again by 3-4 logs) is to the non-cognate A02-GLC (EBV BMLF1) pMHC, rather than the A02-GIL flu peptide. I would strongly encourage the authors to revisit this data and double-check the experiment, because (1) this TCR is just about the most canonical A02-GIL (Flu M158) TCR sequence you can possibly imagine (TRBV19, RS motif, TRAV27/TRAJ42), so it's hard to believe that it would bind so much more weakly than TCR#35-13. And (2) the dextramer data from the 10x experiment shows zero binding to A02-GLC, the supposed tight binder (Decision letter image 1, clone #30, far right red box), and instead strong binding to A02-GIL. And the A02-GLC dextramer did indeed work in the 10x experiment: clone #7 shows strong binding (757/772 cells meet the threshold), and many other clonotypes outside the top 30 also showed specific binding (plus the TCR repertoires for A02-GLC are consistent with previously published repertoire data for this pMHC).

We thank the Reviewer for pointing out this discrepancy.

We have double-checked this experiment and detected a mis-labeling of the peptides displayed for TCR#36-150, which has now been corrected.

Noteworthy, these 2 TCRs #36-150 and #35-13 show very similar behavior. Interestingly, they use different α chains and differ by only two amino acids in the CDR3beta region. As they show remarkably similar quantitative responses, on the one hand, they highlight the robustness of our TCR re-expression and testing technology. On the other hand, they also emphasize the validity of the Levenshtein distance parameter as a proxy of TCRs with similar antigen recognition properties.

This inconsistency between the in vitro data and the 10x dextramer data, like the failure to see A03-KLG binding, should also be pointed out.

We now discuss it.

– In their response, the authors state "we compared the CDR3 length distribution between DP CD3+ cells and CD8 SP cells from our thymic dataset. We did not observe major changes. The distribution and the mean of CDR3 length for the two cell populations remained identical." In fact, the orange distribution is significantly shifted (as the authors show) toward shorter lengths and probably also slightly compressed. It's certainly not the case that the "distribution and mean" remain identical. Perhaps the authors meant to say the "mode"? Still, the distribution is certainly not the same, and seemingly modest shifts in CDR3 length can have significant impacts on neighbor distributions.

Indeed, we should have said that the “medians” are identical, while the means are slightly different.

Shorter CDR3s indeed tend to cluster preferentially. However, the mean varies only by less than one amino acid between the two populations for each donor (shown in red in the Author response image 4), but we are not aware of any study showing that such small shifts in the CDR3 length can have significant impacts on the neighbor distributions.

We have nevertheless revised our sentence in the manuscript regarding the differences in the mean/median distribution.

**Author response image 4. sa2fig4:** 

– In Figure 4C, I wonder whether the authors are taking consistency within expanded clonotypes into account. Is a TCR called as positive for a given dextramer if any cells expressing it are bound by that dextramer, regardless of the fraction those cells represent of the full clonotype?

Yes.

In Decision letter image 1, we can see that even for the specific dextramers, there are occasional cells (typically less than 1 percent of the clone) in the non-binding clonotypes that meet the dextramer UMI threshold. There are all sorts of reasons why we might see this: minor stickiness in the reagents, non-specific dextramer adherence, incomplete washing of non-bound dextramers, incorrect TCR assignment (this happens a fair amount, possibly from ambient TCR transcripts that become trapped in the wrong emulsion).

This was discussed above.

To reiterate, I appreciate that the authors are keeping an open mind about their conclusions.

We are keeping an open mind and are just proposing that what has been (and is) dismissed by many immunologists may be biologically relevant. We hope to have stimulated openness in the reviewer’s appreciation of our interpretations.

I personally feel that the data as presented and as it regards the public 10x dextramer dataset needs to be corrected or revised along with the discussion on this point.

We have drastically revised our presentation and discussion of these results.

Altogether, the important point is that they are functionally polyreactive cells which role deserves attention! Again, how do immunologists explain that you can efficiently fight viral infections with a repertoire of just a few thousand TCRs?